# Single mosquito metatranscriptomics identifies vectors, emerging pathogens and reservoirs in one assay

Joshua Batson[1†], Gytis Dudas[2†], Eric Haas-Stapleton[3†], Amy L Kistler[1†]*, Lucy M Li[1†], Phoenix Logan[1†], Kalani Ratnasiri[4†], Hanna Retallack[5†]

[1]Chan Zuckerberg Biohub, San Francisco, United States; [2]Gothenburg Global Biodiversity Centre, Gothenburg, Sweden; [3]Alameda County Mosquito Abatement District, Hayward, United States; [4]Program in Immunology, Stanford University School of Medicine, Stanford, United States; [5]Department of Biochemistry and Biophysics, University of California San Francisco, San Francisco, United States

**Abstract** Mosquitoes are major infectious disease-carrying vectors. Assessment of current and future risks associated with the mosquito population requires knowledge of the full repertoire of pathogens they carry, including novel viruses, as well as their blood meal sources. Unbiased metatranscriptomic sequencing of individual mosquitoes offers a straightforward, rapid, and quantitative means to acquire this information. Here, we profile 148 diverse wild-caught mosquitoes collected in California and detect sequences from eukaryotes, prokaryotes, 24 known and 46 novel viral species. Importantly, sequencing individuals greatly enhanced the value of the biological information obtained. It allowed us to (a) speciate host mosquito, (b) compute the prevalence of each microbe and recognize a high frequency of viral co-infections, (c) associate animal pathogens with specific blood meal sources, and (d) apply simple co-occurrence methods to recover previously undetected components of highly prevalent segmented viruses. In the context of emerging diseases, where knowledge about vectors, pathogens, and reservoirs is lacking, the approaches described here can provide actionable information for public health surveillance and intervention decisions.

*For correspondence:
amy.kistler@czbiohub.org

[†]These authors contributed equally to this work

Competing interests: The authors declare that no competing interests exist.

## Introduction

Mosquitoes are known to carry more than 20 different eukaryotic, prokaryotic, and viral agents that are pathogenic to humans (*WHO, 2017*). Infections by these mosquito-borne pathogens account for over half a million human deaths per year, millions of disability-adjusted life years (*GBD 2017 Causes of Death Collaborators, 2018*; *GBD 2017 DALYs and HALE Collaborators, 2018*; *GBD 2017 Disease and Injury Incidence and Prevalence Collaborators, 2018*), and periodic die-offs of economically important domesticated animals (*Pagès and Cohnstaedt, 2018*). Moreover, recent studies of global patterns of urbanization and warming, as well as the possibility of mosquito transport via long-range atmospheric wind patterns point to an increasing probability of a global expansion of mosquito habitat and a potential concomitant rise in mosquito-borne diseases within the next two to three decades (*Huestis et al., 2019*; *Kraemer et al., 2019*). While mosquito control has played a major role in eliminating transmission of these diseases in many parts of the world, costs and resources associated with basic control measures, combined with emerging pesticide resistance, pose a growing challenge in maintaining these gains (*Wilson et al., 2020*).

Female mosquitoes take up blood meals from humans and diverse animals in their environment and serve as a major source of trans-species introductions of infectious microbes. For well-studied mosquito-borne human pathogens such as West Nile virus, an understanding of the transmission

dynamics between animal reservoir, mosquito vector, and human hosts has been essential for public health monitoring and intervention (*Hofmeister, 2011*). In contrast, transmission dynamics are less clear for emerging microbes with pathogenic potential. Metagenomic sequencing of individual mosquitoes offers a potential single assay to comprehensively identify mosquito species, the pathogens they carry and the animal hosts that define a transmission cycle.

We also lack a comprehensive understanding of the composition of the endogenous mosquito microbiota, which has been suggested to impact the acquisition, maintenance, and transmission of pathogenic mosquito-borne microbes. For example, *Wolbachia*, a highly prevalent bacterial endosymbiont of insects (*Werren et al., 2008*) has been shown to inhibit replication of various mosquito-borne, human-pathogenic viruses when introduced into susceptible mosquitoes (*Moreira et al., 2009*). These observations have led to the development of *Wolbachia*-based mosquito control programs for *Aedes aegypti* mosquitoes, which vector yellow fever virus, dengue virus, Zika virus, and chikungunya virus. Experimental releases of *Aedes aegypti* mosquitoes transinfected with *Wolbachia* have resulted in a significant reduction in the incidence of dengue virus infections in local human populations. Laboratory-based studies have identified additional endogenous mosquito microbes, such as midgut bacteria and several insect-specific flaviviruses. Greater knowledge of these endogenous microbes could inform their potential use in interfering with mosquito acquisition of and competence to transmit pathogenic *Plasmodium* species and human flaviviruses, respectively. Quantitative analysis of the composition of endogenous microbes and the viruses in individual mosquitoes would be needed to establish a role for these agents in naturally occurring infections and/or transmission of known human pathogens.

Several recent, unbiased metagenomic analyses of batches of mosquito pools collected around the world have begun to address these issues (*Atoni et al., 2018*; *Fauver et al., 2016*; *Frey et al., 2016*; *Li et al., 2015*; *Moreira et al., 2009*; *Pettersson et al., 2019*; *Sadeghi et al., 2018*; *Shi et al., 2019*; *Shi et al., 2015*; *Shi et al., 2017*; *Shi et al., 2016*; *Xia et al., 2018*; *Xiao et al., 2018a*; *Xiao et al., 2018b*) (reviewed in *Atoni et al., 2019*; *Xiao et al., 2018b*). These studies, which have primarily focused on analysis of viruses, have expanded our understanding of the breadth of viral diversity present in mosquito populations worldwide. More recently, similar approaches have been applied to examine the mosquito virome across different life stages of both lab-reared and wild-caught *Aedes albopictus* mosquitoes, providing intriguing insights to the potential stability and diversity of the mosquito virome (*Shi et al., 2020*). Despite these insights, key epidemiologic information needed to direct interventions is still lacking. This includes the measurement of viral prevalence within mosquito populations, their potential reservoir sources, or the impact that additional bacterial and eukaryotic microbes carried by mosquitoes might have on virus carriage, transmission, and pathogenesis.

Single mosquito analyses are required to link blood meal sources, endogenous microbes, and co-occurring pathogens. A handful of small-scale studies have demonstrated that it is possible to identify divergent viruses and evidence of other microbes in single mosquitoes via metagenomic next-generation sequencing (*Bigot et al., 2018*; *Chandler et al., 2015*; *Shi et al., 2019*). Here, we analyzed the metatranscriptomes of 148 individual mosquitoes collected in California, USA. We characterized the composition of their co-infecting microbes, quantified the prevalence and load of detectable viruses and selected bacterial and eukaryotic microbes, and identified blood meal sources and their associated pathogens. Crucially, sequencing a large number of individuals allowed for simple co-occurrence analyses that extended the sensitivity to detect missing or as-yet unidentified viral genome segments with no recognizable homology to previously described sequences. Our findings demonstrate how large-scale single mosquito metatranscriptomics can define both the mosquito's complex microbiota, including mosquito-borne pathogens, and its blood meal sources, thus contributing critical epidemiological information needed to control transmission.

## Results

### Mosquito host speciation by comparative whole transcriptome analysis

Adult *Aedes*, *Culex*, and *Culiseta* mosquito species circulating in California in late fall of 2017 were collected to acquire a diverse and representative set of 148 mosquitoes for metatranscriptomic next-generation sequencing (mNGS) analysis. We targeted collections across a variety of habitats

within five geographically distinct counties in Northern and Southern California (*Figure 1—figure supplement 1*). Visual mosquito species identification was performed at the time of collection (Materials and methods, results are summarized in *Figure 1—source data 1*). Primarily female mosquitoes were included to enrich for blood-feeding members of the population responsible for transmission of animal and human diseases. Total RNA extracted from each mosquito was used as the input template for mNGS to capture both polyadenylated and non-polyadenylated host, viral, prokaryotic, and eukaryotic RNAs (Materials and methods; overall sequence yields for each mosquito are summarized in *Figure 1—source data 2*).

Given the important role of accurate identification of mosquito species for understanding geospatial mosquito circulation and vector-pathogen interactions, and the potential for human error in visual inspection, we investigated if single mosquito mNGS could provide a complementary, unbiased molecular method for identifying mosquito species. Because complete genome sequences were not available for all mosquito species identified visually in this set, we applied a reference-free, kmer-based approach (*Harris, 2018*) to compute pairwise genetic distances between the complete metatranscriptomes acquired for each of the 148 mosquitoes. Samples were grouped using hierarchical clustering and the most common visually identified species within each group was taken as a consensus species call for that group (*Figure 1*, and see *Figure 1—figure supplement 2* for detailed alignment of visual calls with the clustered genetic distance matrix, *Figure 1—source data 1*, and *Figure 1—figure supplement 2—source data 1* for underlying data). These molecular groupings of mosquito genera and species agreed the visual calls for 95% of the specimens (n = 140/147, one sample had no visual identification). The discordant calls occurred in two contexts reported to present challenges to morphology-based speciation: (1) within the *Culex* genus in which genetic hybridization among species members has been documented and reported to confound accurate morphological speciation in California *Cornel et al., 2003*; *Cornel et al., 2003*; *Kothera et al., 2012*; *McAbee et al., 2008*; and (2) between samples belonging to the *Culex* and *Culiseta* genera that share some overlap in morphology, and require detection of features (perspiracular bristles and subcostal wing vein bristles) that can be lost or damaged during trapping and handling (*Darsie and Ward, 2016*). Thus, we used the transcriptome-based species calls for this study. There is additional

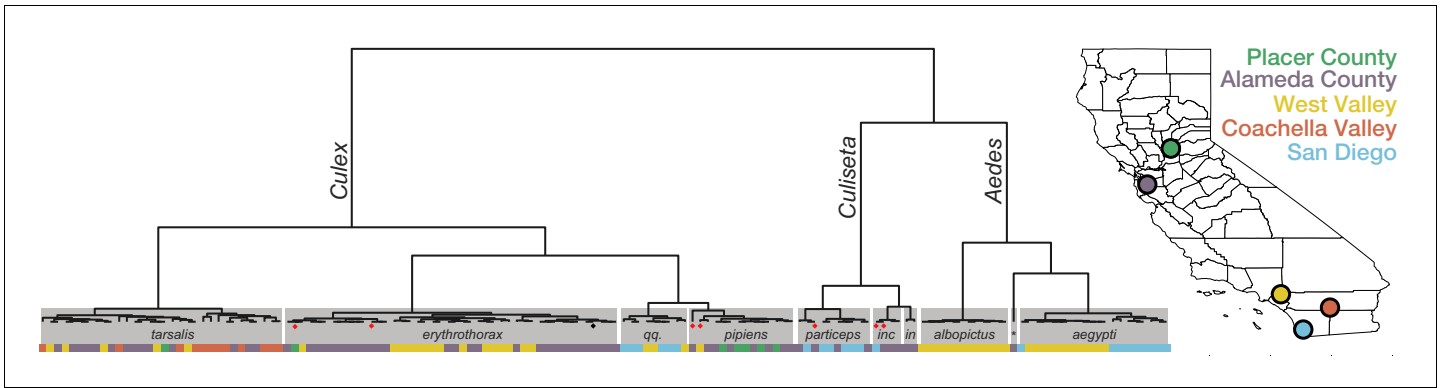

**Figure 1.** Whole transcriptome analysis for mosquito species identification. Hierarchical clustering of pairwise single-nucleotide polymorphism (SNP) distances between whole transcriptome sequences from the 148 mosquitoes included in this study estimated using SKA (*Harris, 2018*). The inferred mosquito species for each cluster (text in gray boxes) is the consensus of the species calls made by visual inspection during sample collection for samples in that cluster (*qq.* = *Culex quinquefasciatus*, *particeps* = *Culiseta particeps*, *inc* = *Culiseta incidens*, *in* = *Culiseta inornata*, *albo* = *Aedes albopictus*, *=Aedes dorsalis*). Red dots below the nodes on the tree highlight mismatches (n = 7) between consensus transcriptome species call and initial visual species call; black dot indicates a sample missing a visual species call. Color bar below the tree shows the collection location for each sample, coded according to the California map legend at right.

The online version of this article includes the following source data and figure supplement(s) for figure 1:

**Source data 1.** Mosquito demographic metadata.

**Source data 2.** Per sample sequence read yield metadata.

**Figure supplement 1.** Diversity of individual mosquitoes collected across California.

**Figure supplement 2.** Transcriptome-based identification of mosquito species.

**Figure supplement 2—source data 1.** SNP distance matrix data underlying *Figure 1—figure supplement 2*.

within-species structure visible on the hierarchical clustering tree that, for *Aedes aegypti* and *Culex erythrothorax*, coincides with geographic structure, raising the possibility that molecular methods may also provide insight into the distribution of subspecies as well (*Figure 1*). Taken together, these data show that comparative transcriptome analysis of single mosquito mNGS data can provide critical information regarding the identity and diversity of circulating mosquitoes.

## Comprehensive and quantitative analysis of non-host sequences detected in single mosquitoes

To understand the distribution of species within the microbial cargo of the mosquitoes, we first examined the overall proportion of non-host reads assembled into contigs that could be assigned to viral, bacterial, and eukaryotic taxa. A detailed overview of the analysis we applied to identify, assemble, classify, and quantify all the non-host contigs and their associated read counts is provided in Materials and methods (graphically summarized in *Figure 2—figure supplement 1*, and *Figure 2—figure supplement 1—source data 1*; mosquito reference sequences provided in *Figure 2—source data 1*). Details of the per mosquito breakdown of non-host read assignment across high level taxonomic categories are provided in *Figure 2—figure supplement 2*, and *Figure 2—figure supplement 2—source data 1*. *Figure 2* provides a quantitative treemap overview of how the assembled non-host reads mapped across the viral, prokaryotic, and eukaryotic taxa (see *Figure 2—figure supplement 3* for a higher resolution treemap view, and *Figure 2—figure supplement 3—source data 1* for underlying data). In sum, we were able to classify, to at least kingdom level, 77% of the 21.8 million non-host reads that assembled into contigs with more than two reads.

## Diverse known and novel RNA virus taxa dominate the mosquito microbiota

We found that the vast majority of the non-host reads that assembled into contigs corresponded to complete viral genomes (10.9 million reads of the 13 million total non-host reads assembled into contigs; *Figure 2*, all blocks in the treemap annotated with suffix '-viridae'). Positive-sense single-stranded RNA viruses made up the most abundant class of detected viruses (7.4 million reads of the 10.9 million viral reads; *Figure 2* blocks labeled *Solemoviridae, Luteoviridae, Tombusviridae, Narnaviridae, Flaviviridae, Virgaviridae, and Iflaviridae*), negative-sense single-stranded RNA viruses made up the next most abundant virus category (2.25 million reads of the 10.9 million viral reads; *Figure 2* blocks labeled *Peribunayviridae, Phasmaviridae, Phenuiviridae, Orthomyxoviridae, Chuviridae, Rhabdoviridae, and Ximnoviridae*), and double-stranded RNA viruses formed the third most abundant virus category (0.94 million reads of the 10.9 million viral reads; *Figure 2* blocks labeled *Chrysoviridae, Totiviridae, Partitiviridae, and Reoviridae*). In many cases, multiple independent isolates of complete viral genomes were recovered across the individual mosquito specimens. In all, a total of 70 distinct viral taxa were recovered, 46 of which correspond to distinctly divergent novel viruses (*Table 1*). Intriguingly, only 10 of the 24 previously described viral taxa have been recovered from mosquitoes in California (*Table 1*, rows highlighted in gray; *Chandler et al., 2015*; *Sadeghi et al., 2018*). We cannot rule out that the known and novel viral species that correspond to viral families previously thought to only infect plants and fungi, (e.g. the *Chrysoviridae, Totiviridae,* Luteoviridae, *and Solemoviridae, Table 1*) could potentially be explained by environmental exposures retained on the surface of the mosquito. However, emerging evidence from mosquito and other insect metatranscriptomic studies has indicated that these viral taxa are tightly associated with, if not actually infecting, mosquitoes and other insects (*Shi et al., 2017*; *Shi et al., 2016*; *Xiao et al., 2018b*).

The balance of additional reads that could be assigned to viral taxa corresponded to reads assembled into contigs that were clearly viral in origin but incomplete by either not associated with an RNA-dependent polymerase (0.37 million reads, *Figure 2*, light gray block labeled 'uncurated viruses') or associated with contigs aligning to viral taxa that were detected at levels too low to visualize on the treemap. This latter set of diverse viral taxa corresponded to several types of DNA viruses, such as nucleocytoplasmic large DNA viruses, members of the *Polydnaviridae, Alphabaculovirus, Nudiviridae,* and *Circovirus-like* sequences, and phages (data not shown). Some of these viral taxa likely reflect *bona fide* infections, while others are likely the result of indirect infections. For example, six distinct types of *Botourmiaviridae*, a family of viruses primarily known to infect fungi,

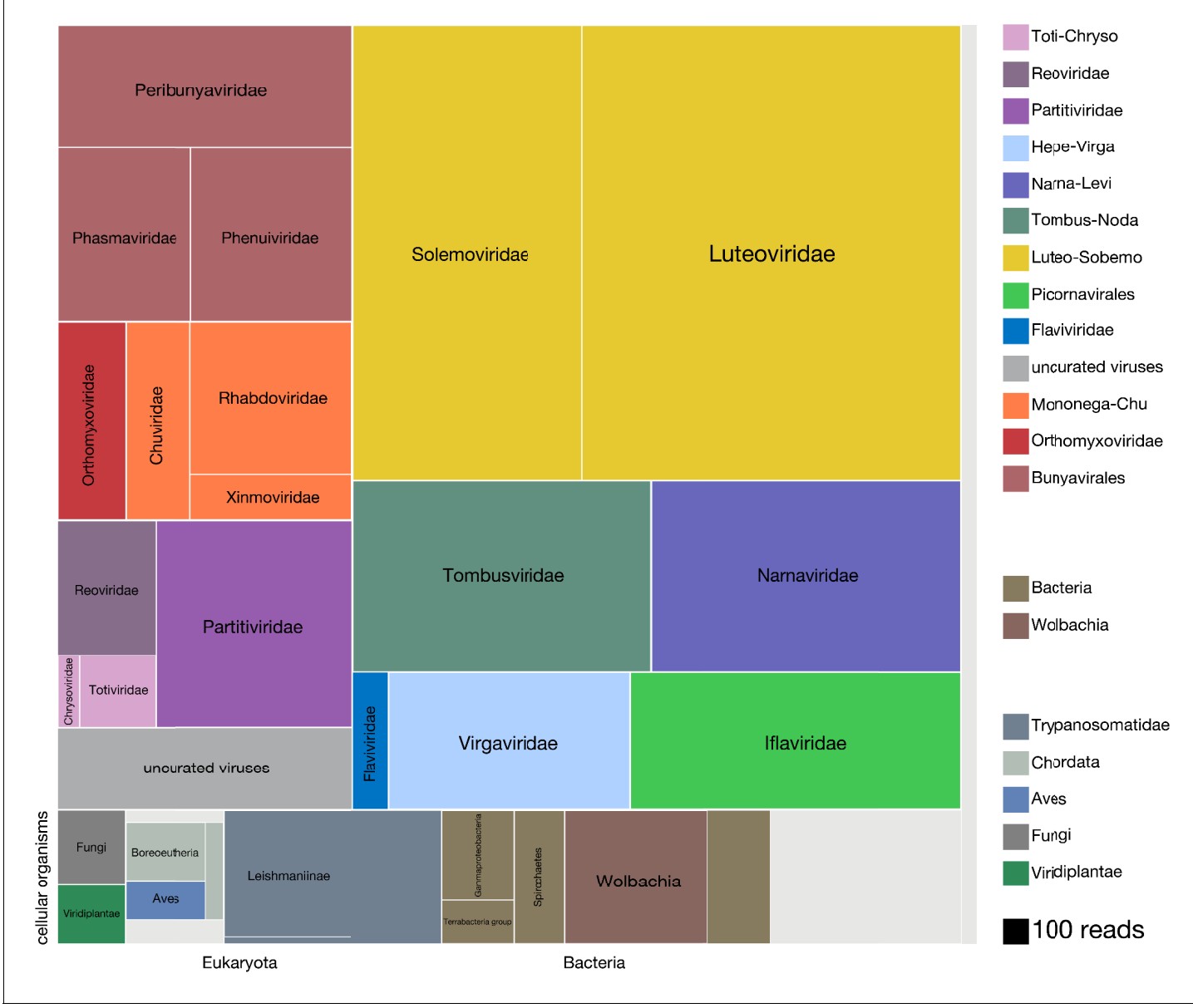

**Figure 2.** Viruses dominate the microbial signature of mosquitoes. Treemap plot of the proportion of reads recovered among non-host contigs assembled from the 148 single mosquitoes, with viral taxa making up the top portion of the plot above 'cellular organisms' label at lower left edge of image, which designates reads assembled into contigs encompassing prokaryotic taxa, the eukaryotic taxa and the taxonomically ambiguous 'cellular organisms' that were not possible to assign to a higher resolution (light gray box). The block areas are plotted in proportion to the number of reads assembled into contigs that could be assigned to a given taxon (see area scale, legend). The gray block labeled 'uncurated viruses' corresponds to the number of reads assembled into contigs that were clearly viral in origin, but difficult to further resolve due to fragmented genomes and/or the lack of an associated RNA-dependent RNA polymerase.

The online version of this article includes the following source data and figure supplement(s) for figure 2:

**Source data 1.** Detailed summary of sequences used for mosquito host reference.

**Figure supplement 1.** Summary of single mosquito mNGS analysis pipeline with sequence recovery yields.

**Figure supplement 1—source data 1.** Data underlying counts summarized in *Figure 2—figure supplement 1*.

**Figure supplement 2.** Summary of all reads in each mosquito.

**Figure supplement 2—source data 1.** Data underlying *Figure 2—figure supplement 2*.

**Figure supplement 3.** High-resolution breakdown the microbial signature of mosquitoes.

**Figure supplement 3—source data 1.** Data underlying *Figure 2*, and *Figure 2—figure supplement 3*.

**Figure supplement 4.** Analysis of novel peribunya-like virus showing completes of genome recovery.

**Figure supplement 4—source data 1.** Underlying data for *Figure 2—figure supplement 4*.

**Table 1.** Complete genomes of known and novel viral taxa recovered in this study*.

| Genome type | Viral family | Virus name | Novel? | Number detected overall | Number in *Aedes* samples | Number in *Culex* samples | Number in Culiseta samples |
|---|---|---|---|---|---|---|---|
| Single-stranded positive sense RNA | *Botourmiaviridae* | Patollo virus | TRUE | 1 | 0 | 1 | 0 |
| | *Botourmiaviridae* | Picullus virus | TRUE | 1 | 0 | 1 | 0 |
| | *Botourmiaviridae* | Poccolus virus | TRUE | 1 | 0 | 1 | 0 |
| | *Botourmiaviridae* | Pikulas virus | TRUE | 1 | 0 | 1 | 0 |
| | *Botourmiaviridae* | Pecols virus | TRUE | 1 | 0 | 1 | 0 |
| | *Botourmiaviridae* | Patulas virus | TRUE | 1 | 0 | 1 | 0 |
| | *Dicistroviridae* | Wuhan insect virus 33 | FALSE | 3 | 3 | 0 | 0 |
| | *Flaviviridae* | Placeda virus | TRUE | 3 | 0 | 3 | 0 |
| | *Flaviviridae* | Culex flavivirus | FALSE | 3 | 0 | 3 | 0 |
| | *Flaviviridae* | Calbertado virus | FALSE | 1 | 0 | 1 | 0 |
| | *Iflaviridae* | Culex iflavi-like virus 4 | FALSE | 6 | 2 | 4 | 0 |
| | *Iflaviridae* | Calumiyane virus | TRUE | 3 | 0 | 0 | 3 |
| | *Iflaviridae* | Culex iflavi-like virus 3 | FALSE | 2 | 0 | 2 | 0 |
| | *Iflaviridae* | Calfluga virus | TRUE | 1 | 0 | 0 | 1 |
| | *Leviviridae* | Chimba virus | TRUE | 1 | 0 | 1 | 0 |
| | *Leviviridae* | Ulae virus | TRUE | 1 | 1 | 0 | 1 |
| | *Luteoviridae* | Culex-associated Luteo-like virus | FALSE | 7 | 0 | 7 | 0 |
| | *Luteoviridae* | Geolu virus | TRUE | 1 | 0 | 0 | 1 |
| | *Narnaviridae* | **Culex narnavirus 1** | **FALSE** | **43** | **0** | **43** | **0** |
| | *Narnaviridae* | Whakaata virus | TRUE | 1 | 0 | 0 | 1 |
| | *Solemoviridae* | Marma virus | FALSE | 39 | 0 | 39 | 0 |
| | *Solemoviridae* | **Culex mosquito virus 6** | **FALSE** | **15** | **0** | **15** | **0** |
| | *Solemoviridae* | Guadeloupe mosquito virus | FALSE | 8 | 8 | 0 | 0 |
| | *Solemoviridae* | Wenzhou sobemo-like virus 4 | FALSE | 7 | 7 | 0 | 0 |
| | *Solemoviridae* | Kellev virus | TRUE | 2 | 0 | 0 | 2 |
| | *Tombusviridae* | Hubei mosquito virus 4 | FALSE | 18 | 0 | 18 | 0 |
| | *Tombusviridae* | Erebo virus | TRUE | 4 | 0 | 3 | 1 |
| | *Tombusviridae* | Vai augu virus | TRUE | 2 | 0 | 2 | 0 |
| | *Tymoviridae* | Guadeloupe Culex tymo-like virus | FALSE | 1 | 0 | 1 | 0 |
| | *Virgaviridae* | Hubei virga-like virus 2 | FALSE | 36 | 0 | 36 | 0 |
| | *Virgaviridae* | Culex pipiens-associated Tunisia virus | FALSE | 13 | 0 | 13 | 0 |

*Table 1 continued on next page*

*Table 1 continued*

| Genome type | Viral family | Virus name | Novel? | Number detected overall | Number in *Aedes* samples | Number in *Culex* samples | Number in Culiseta samples |
|---|---|---|---|---|---|---|---|
| Single-stranded negative sense RNA | *Chuviridae* | Culex mosquito virus 4 | FALSE | 6 | 0 | 6 | 0 |
| | *Orthomyxoviridae* | Üsinis virus | TRUE | 15 | 15 | 0 | 0 |
| | *Orthomyxoviridae* | Wuhan mosquito virus 6 | FALSE | 13 | 0 | 13 | 0 |
| | *Orthomyxoviridae* | Guadeloupe mosquito quaranja-like virus 1 | FALSE | 5 | 5 | 0 | 0 |
| | *Orthomyxoviridae* | Astopletus virus | TRUE | 2 | 0 | 2 | 0 |
| | *Peribunyaviridae* | Culex bunyavirus 2 | FALSE | 30 | 0 | 30 | 0 |
| | *Peribunyaviridae* | Udune virus | TRUE | 1 | 0 | 1 | 0 |
| | *Peribunyaviridae* | Dumanli virus | TRUE | 1 | 0 | 1 | 0 |
| | *Phasmaviridae* | Barstukas virus | TRUE | 16 | 16 | 0 | 0 |
| | *Phasmaviridae* | Miglotas virus | TRUE | 16 | 0 | 16 | 0 |
| | *Phenuiviridae* | Niwlog virus | TRUE | 8 | 0 | 8 | 0 |
| | *Rhabdoviridae* | Merida virus | FALSE | 9 | 4 | 5 | 0 |
| | *Rhabdoviridae* | Stang virus | TRUE | 4 | 0 | 4 | 0 |
| | *Rhabdoviridae* | Elisy virus | TRUE | 3 | 0 | 3 | 0 |
| | *Rhabdoviridae* | Canya virus | TRUE | 2 | 0 | 2 | 0 |
| | *Xinmoviridae* | Gordis virus | TRUE | 9 | 0 | 0 | 9 |
| | *Xinmoviridae* | Aedes anphevirus | FALSE | 2 | 2 | 0 | 0 |
| Double-stranded RNA | *Chrysoviridae* | Hubei chryso-like virus 1 | FALSE | 2 | 0 | 2 | 0 |
| | *Chrysoviridae* | Keturi virus | TRUE | 1 | 0 | 1 | 0 |
| | *Partitiviridae* | Netjeret virus | TRUE | 11 | 11 | 0 | 0 |
| | *Partitiviridae* | Nefer virus | TRUE | 10 | 0 | 2 | 8 |
| | *Partitiviridae* | Nebet virus | TRUE | 2 | 0 | 0 | 2 |
| | *Reoviridae* | Elemess virus | TRUE | 6 | 0 | 6 | 0 |
| | *Reoviridae* | Lasigmu virus | TRUE | 1 | 1 | 0 | 0 |
| | *Reoviridae* | Lobuck virus | TRUE | 1 | 1 | 0 | 0 |
| | *Totiviridae* | Tzifr virus | TRUE | 27 | 0 | 27 | 0 |
| | *Totiviridae* | Lotchka virus | TRUE | 19 | 0 | 19 | 0 |
| | *Totiviridae* | Mika virus | TRUE | 10 | 0 | 10 | 0 |
| | *Totiviridae* | Aedes aegypti totivirus | FALSE | 5 | 5 | 0 | 0 |
| | *Totiviridae* | Snelk virus | TRUE | 4 | 0 | 4 | 0 |
| | *Totiviridae* | Gouley virus | TRUE | 4 | 0 | 4 | 0 |
| | *Totiviridae* | Stinn virus | TRUE | 2 | 0 | 2 | 0 |
| | *Totiviridae* | Hagerguy virus | TRUE | 2 | 0 | 0 | 2 |
| | *Totiviridae* | Nuyav virus | TRUE | 1 | 1 | 0 | 0 |
| | *Totiviridae* | Mughataa virus | TRUE | 1 | 0 | 0 | 1 |
| | *Totiviridae* | Koroku virus | TRUE | 1 | 0 | 0 | 1 |
| | *Totiviridae* | Emileo virus | TRUE | 1 | 0 | 0 | 1 |
| | *Totiviridae* | Gissa virus | TRUE | 1 | 0 | 0 | 1 |
| | *Totiviridae* | Totivirus-like Culex mosquito virus 1 | FALSE | 1 | 0 | 1 | 0 |

[*]Bold text rows highlight viruses previously detected in California (***Chandler et al., 2015*** and ***Sadeghi et al., 2018***).

The online version of this article includes the following source data for Table 1:

Source data 1. Detailed information on the 70 viral genomes recovered in this study.

Source data 2. Blast alignment information on each of the isolates of the 70 viral genomes identified by homology.

were all detected at very low levels in a single mosquito in which the ergot fungus, *Claviceps*, a more likely *Botourmiaviridae* host species, was also detected.

### *Trypanosomatidae* and vertebrate species are major constituents of the eukaryotic taxa

An additional 2.2 million of the 13 million non-host reads assembled into contigs that mapped to non-viral taxa. Just under 1 million reads could be assigned to eukaryotic taxa (0.8 million reads total, *Figure 2*, bottom left row of boxes). Members of *Trypanosomatidae* comprised more than 50% of these reads (0.45M reads), with a significant fraction assigned to the subfamily *Leishmaniinae*, that encompasses multiple *Trypanosomatidae* species known to infect insects and vertebrates. The second most abundant group of eukaryotes detected in the dataset were *Bilateria* (animals) with 0.20 million reads corresponding to mammals (*Boreoeutheria*, 73,000 reads) and birds (*Aves*, 51,000 reads), followed by invertebrates (*Ecdysozoa*, 36,000 reads (not shown); see *Figure 2—figure supplement 3*, and *Figure 2—figure supplement 3—source data 1* for higher resolution details of these and other notable lower abundance eukaryotic taxa detected). The reads derived from vertebrate taxa almost certainly belong to blood meal hosts, which we investigate in detail below. Fungal and plant contigs made up the remainder of the eukaryotic reads we captured from individual mosquito sequencing, with 79,000 and 62,000 total reads, respectively.

### *Wolbachia* species make up the majority of prokaryotic taxa

Prokaryotic contigs encompassed 0.7 million non-host reads. Among the prokaryotic taxa detected, *Wolbachia*, a known endosymbiont of *Culex quinquefasciatus* (*Werren et al., 2008*), comprised most of the reads (0.22 million reads, *Figure 2*, bottom central row of brown-hued blocks). Various other bacterial taxa were detected at lower abundance; that is members of *Alphaproteobacteria*, *Gammaproteobacteria*, *Terrabacteria* group, and *Spirochaetes* (*Spironema culicis* (73,000 reads), a bacterial species previously detected in *Culex* mosquitoes (*Cechová et al., 2004*; *Duguma et al., 2019*), makes up 68% of the *Spirochaetes* reads). A higher resolution overview of the lowest common ancestor (LCA) species we could assign within each of these four broad categories is provided in *Figure 2—figure supplement 3*, *Figure 2—figure supplement 3—source data 1*, and *Figure 3—source data 2*. Interestingly, these results largely agreed with data obtained for the *Culex* and *Aedes* species in prior sequencing studies involving more directed capture of prokaryotic and eukaryotic taxa via 16S rRNA metabarcoding of mosquitoes collected in Thailand (*Thongsripong et al., 2018*).

### Ambiguous and metagenomic 'dark matter' sequences are present

A significant portion of the non-host reads assembled into contigs with sequences that were taxonomically ambiguous. Approximately 0.55 million reads assembled into contigs with a lowest common ancestor (LCA) assigned to the taxonomy nodes of 'root' or 'cellular organisms' (*Figure 2*, unlabeled light gray box). A much larger fraction of non-host reads – approximately 4.7 million reads – corresponded to metagenomic 'dark matter', that is contigs without any recognizable sequence homology to previously published sequences. Contig co-occurrence analysis across the individual mosquito sequence results (see main text below) allowed us to identify additional viral contigs from this set of contigs, contributing 0.34 million reads to the total tally of detectable viral reads in the mosquito microbiota.

Together these data establish the utility of our comprehensive single mosquito mNGS analyses to define the composition and diversity of the mosquito metatranscriptome. The sensitivity of our analysis reveals endogenous constituents of the mosquito microbiome, the source of their blood meals, and the potential human and animal pathogens they carry, and even viruses that selectively infect mosquito-associated fungi.

### Identifying constituents of the mosquito microbiota

To define components of the mosquito microbiota and investigate the potential variation between individuals, we next analyzed the distribution of the viral, prokaryotic, and microbial eukaryotic taxa detectable within the non-host compartment of individual mosquitoes. For this analysis, we focused on the non-host reads assembled into contigs among the individual mosquitoes. To estimate the

composition and proportions of microbial agents detectable within each mosquito, the fraction of non-host reads aligning to each contig corresponding to a viral, bacterial, or eukaryotic microbe sequence was computed. (A broader view of the per-mosquito taxonomic breakdown is provided in *Figure 2—figure supplement 2*, and *Figure 2—figure supplement 2—source data 1*) In *Figure 3*, we provide a higher resolution view of the diverse viral taxa and selected bacterial agents that were detectable at a level above 1% as bars. Contigs that were supported by <1% of the non-host reads assembled into contigs are also included and plotted as dark circle symbols above the x-axis coordinate for the *Wolbachia* panel in *Figure 3*, and gray bars in the *Trypanosomatidae*, *Apicomplexa*, and *Nemotoda* panels at the base of *Figure 3*. Taken together, these data reveal unprecedented insight into the heterogeneity of the microbiota associated within individuals, and between mosquito species and collection sites.

## Viral diversity and prevalence measured in single mosquitoes

Given their predominance and potential relevance to human disease, we first examined the 70 unique viral taxa we detected on a single mosquito basis (*Table 1*, *Table 1—source data 1*). Of these, only 24 were closely related or identical to previously identified mosquito viruses. The remaining 46 viral genomes shared less than 85% amino acid sequence identity to any publicly available viral sequences (See 'Virus' contigs in *Table 1—source data 2*). Despite this divergence, family-level

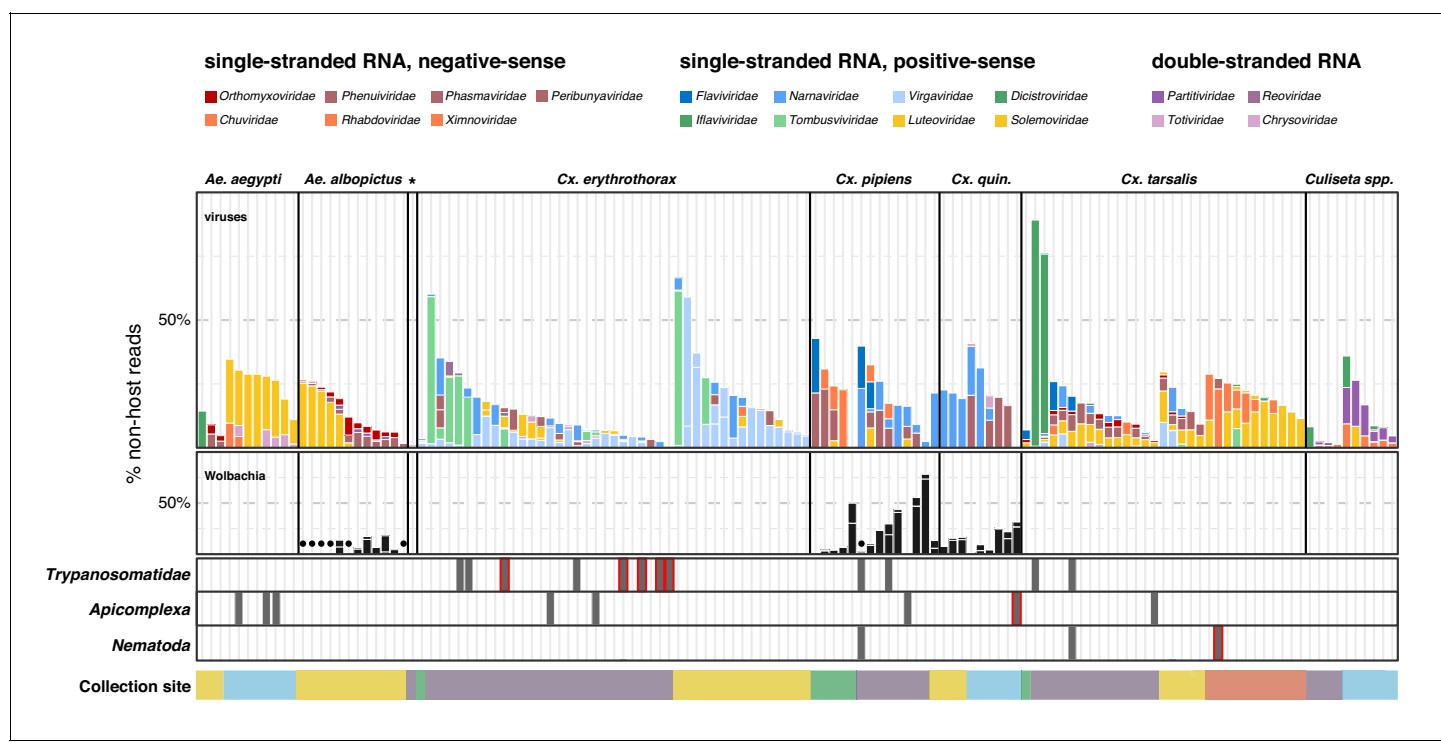

**Figure 3.** Single mosquito sequencing reveals the heterogeneity of microbiota between individuals, species and locations. Plotted bars report the proportions of non-host reads from individual mosquitoes that are supported by ≥1% of non-host reads corresponding to assembled contigs identified as viruses (top panel), *Wolbachia* (middle panel), and selected eukaryotic microbes (bottom panel). Species are color coded as indicated in the legend (top of graph). Plotted symbols on the *Wolbachia* and eukaryotic panels indicate microbes confidently identified, but present at <1% of non-host reads middle panel, black circles = *Wolbachia* taxa; bottom panels, gray bars are plotted in samples with *Trypanosomatidae*, *Apicomplexa* or *Nematoda* taxa; gray bars outlined in red indicate detection of these taxa at ≥1% of non-host reads. Samples were clustered by mosquito species (top labels) and ordered by: (i) collection site location from north to south (indicated at the bottom, colored as in *Figure 1*) and (ii) viral abundance (descending order, left to right).

The online version of this article includes the following source data and figure supplement(s) for figure 3:

**Source data 1.** Underlying virus data for *Figure 3*.

**Source data 2.** Underlying bacteria data for *Figure 3*.

**Source data 3.** Underlying eukaryotic data for *Figure 3*.

**Figure supplement 1.** Evidence for a potential interplay between Wolbachia and viral infections.

sequence conservation of genomic features allowed us to confirm complete genome sequence recovery. For example, conserved sequences were identified at the 5' and 3' ends of bunyavirus segments in a novel peribunya-like virus (*Table 1*, Udune virus; *Figure 2—figure supplement 4*, and *Figure 2—figure supplement 4—source data 1*) with 28% to 76% amino acid identity to its closest relative. Only a single mosquito harbored this virus, and at a relatively low abundance of reads (0.02% of total reads). Combining this approach with the co-occurrence investigation detailed below (Figure 6 and main text), we were able to identify and assign the 45 additional novel genomes to specific viral taxa. Thus, single mosquito analysis provides a highly sensitive approach to detect new and divergent viral species circulating in mosquito populations, even when present at low prevalence or abundance.

Single mosquito analysis also shed light on the variability in the composition and number of viral reads both within and across mosquito species (*Figure 3*, top panel), and their corresponding collection sites (indicated by colored bar plot at the bottom of the plots). Importantly, among single mosquitoes, co-infections predominated, with 88% of mosquitoes harboring two or more (median 3) distinct viral taxa (*Figure 4*, panel A; *Figure 4—source data 1*). Focused analysis of viral species within single mosquitoes provides the opportunity to examine the proportion of viruses within each of these co-infections, which in turn can inform and extend our understanding of the distribution patterns of known and emerging novel viruses within the mosquito population, and the frequency of associated co-infecting viruses. *Figure 3*, shows the wide range in the number and type of viruses that are detected across individual mosquitoes. For instance, several *Culex* mosquito species stand out as outliers harboring only a single viral species encompassing a large proportion of the non-host reads assembled into contigs in that mosquito (*Figure 3*, top panel: *Iflaviridae* species [dark green bars] in *Culex tarsalis*; *Tombusviridae* and *Virgaviridae* species [light blue bars] in *Culex erythrothorax*). At the other end of the spectrum are multiple examples of individual mosquitoes that do not stand out with regard to the proportion of non-host reads assembled into viral contigs, yet still harbor a mixture of 4 or more viruses present at >1% of the non-host reads assembled into contigs (*Figure 3*, top panel - see especially *Culex tarsalis* and *Culiseta* species plots). Other viruses are detected broadly across diverse mosquito species (*Figure 3*, top panel, see the *Solemoviridae/Luteoviridae* [yellow bars], *Narnaviridae* [blue bars], *Virgaviridae* [light blue bars], and *Dicistroviridae/Iflaviridae* [dark green bars]). Interestingly, in some mosquito species, these viruses are the predominant proportion of the non-host reads assembled into viral contigs, while in other mosquito species where these viruses are detected, they make up only a minor fraction. For example, compare *Solemoviridae/Luteoviridae* [yellow bars] in *Aedes* and *Culex tarsalis* species, or *Virgaviridae* [light blue bars] in *Culex erythrothorax* and *Culex tarsalis* species. The opposite pattern of virus distribution where viral species are restricted to a single mosquito species, for example the *Partitiviridae and Reoviridae* among the *Culiseta* species (*Figure 3*, top panel, right edge of plot, dark purple bars). These distinct patterns of viral distribution point to potentially testable hypotheses as to their causes, such as mosquito species susceptibility or competence to vector a virus, the potential pathogenicity of a given virus (or mixture of viruses), or factors in the environment, such as food sources or weather. Regardless of the ultimate source of this variability, such insights are only possible by analyzing mosquitoes individually rather than in bulk.

This variation is particularly relevant when we consider that viral abundance is often calculated based on bulk mosquito sequencing, which does not provide information about the prevalence or heterogeneity in abundance of a virus across the mosquito population. Importantly, we find that the average abundance of a virus (*i.e.* the average number of reads across a set of mosquitoes) is not necessarily predictive of the prevalence of that virus (*i.e.* the number of mosquitoes in which it occurs). For example, Culex narnavirus one and Culex pipiens-associated Tunisia viruses were found at similar abundance in *Culex erythrothorax* mosquitoes obtained from the same collection site in West Valley; however, the latter was three times more prevalent (30% vs 90%, *Figure 4*, panel B; *Figure 4—source data 2*). A more global view of viral diversity and prevalence across mosquito species is shown in (*Figure 4*, panel C; *Figure 4—source data 3*) which plots the fraction of individuals infected for each mosquito species and each virus detected in our study. This quantitative and comprehensive analysis of the prevalence of mosquito-borne viruses would not be possible without single mosquito sequencing, yet provides critical epidemiological information needed to manage the transmission of mosquito-borne viruses. For example, the sampled mosquito genera (*Culex*, *Aedes*, and *Culiseta*) have distinct viromes, with only four viruses shared across genus boundaries and even

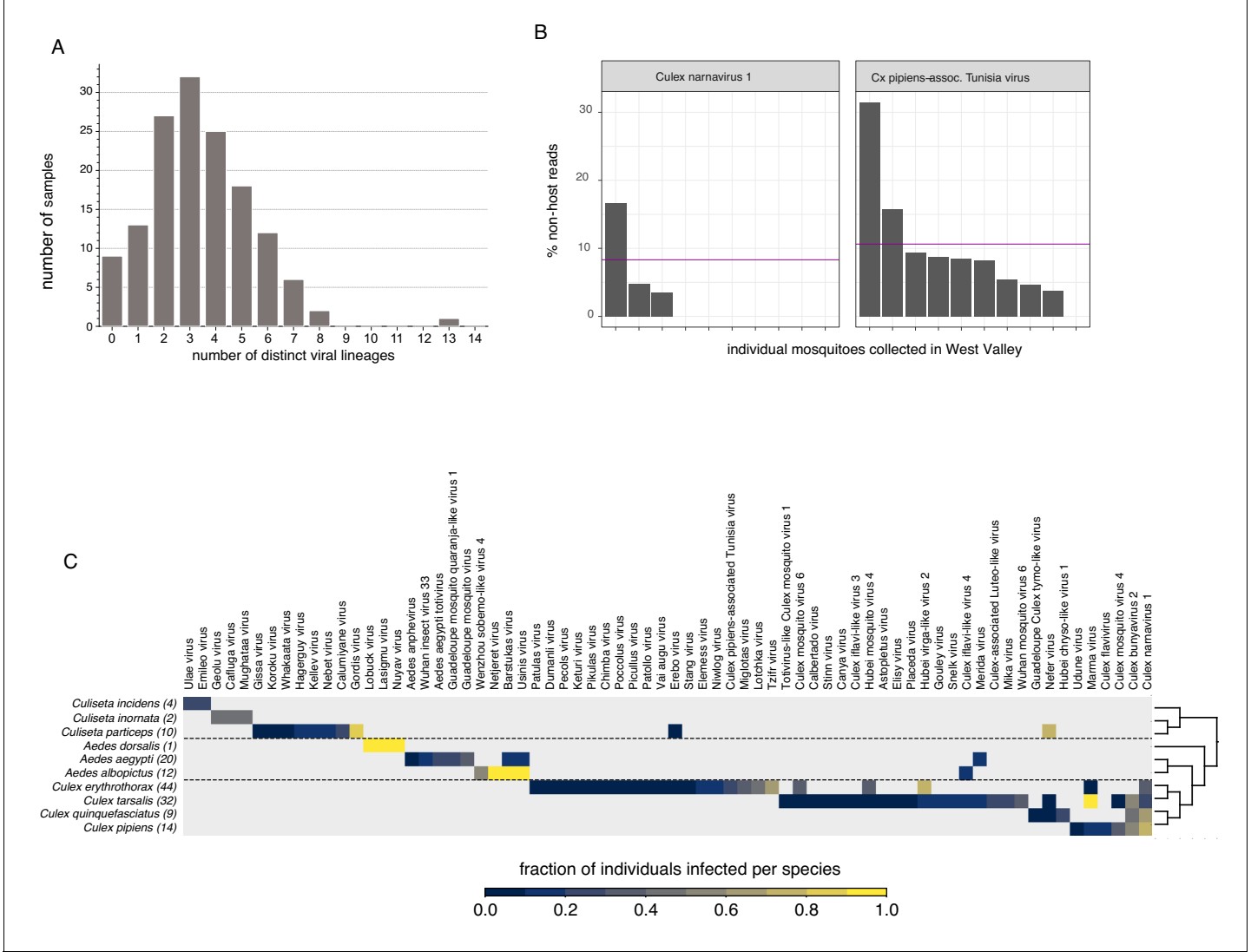

**Figure 4.** Quantifying viral diversity in and prevalence among single mosquitoes. (A) Distribution of mosquitoes within the study in which no, one, or multiple viral lineages were detectable. The sample with 13 distinct viral lineages is sample CMS002_053a that contains six Botourmia-like viruses thought to primarily infect fungi and in which evidence of an ergot fungus was detected. (B) An example of viruses with similar bulk abundance, but different prevalence. Both Culex narnavirus one and Culex pipiens-associated Tunisia virus were found among the 10 *Culex erythrothorax* mosquitoes collected at the same collection site in West Valley. The bulk abundances were calculated by the mean % non-host reads averaged across the 10 mosquitoes for Culex narnavirus one and Culex pipiens-associated Tunisia virus were 8.3% and 10.6%, respectively (as indicated by the red line). However, the prevalence (i.e. the percent of single mosquitos carrying the virus) was markedly different (C). Global analysis of viral prevalence measured in this study. For each virus, the fraction of individuals infected within each species was calculated, shown on a color scale. Mosquito species arranged according to a phylogeny based on the cytochrome c oxidase subunit I (COI) gene.

The online version of this article includes the following source data for figure 4:

**Source data 1.** Data underlying *Figure 4*, panel A (coinfection plot).
**Source data 2.** Data underlying *Figure 4*, panel B (prevalence-abundance plot).
**Source data 3.** Data underlying *Figure 4*, panel C (mosquito - virus species plot).

then, only Merida virus (-ssRNA) and Culex iflavi-like virus 4 (+ssRNA) are shared by *Aedes* and *Culex* mosquitoes. Within each genus, viruses appear to be largely unique to species, although some overlap is detectable (*Figure 4*, panel C; *Figure 4—source data 3*), potentially reflecting greater similarities in ecology and physiology (*Longdon et al., 2014*) that enable an easier flow of viruses between populations.

## Exploring the impact of *Wolbachia* endosymbionts

We restricted our single mosquito analysis of detectable prokaryotes to *Wolbachia* given its abundance and evidence suggesting that as an endosymbiont it could impact the microbiota of its mosquito hosts. *Wolbachia* was detected in 32 mosquitoes belonging to *Culex quinquefasciatus*, *Culex pipiens*, and *Aedes albopictus* species (*Figure 3*, middle panel, black bars and circle symbols). These observations are consistent with previous reports of wild-caught mosquito species that are naturally infected with *Wolbachia* (*Kittayapong et al., 2000*; *Rasgon and Scott, 2004*). Among these three species, *Wolbachia* was detected in all or nearly all of the mosquitoes. Thus, it was not possible to draw definitive conclusions regarding whether the presence or absence of *Wolbachia* influenced the composition of detectable co-occurring viral taxa among these mosquito species. However, the fraction of non-host reads assembled into contigs that were assigned to *Wolbachia* varied dramatically among the individual mosquitoes, from <1% to as high as 74% (*Figure 3*, middle panel, black circles and black bar plots, respectively), and revealed interesting trends that would require further validation. For example, for *Ae. albopictus*, individuals with higher levels of detectable *Wolbachia* (*Figure 3*, central panel: samples with black bars) exhibited shift in viral species, with a lower proportion of positive-sense RNA viruses (*Solemonviridae and Luteoviridae*) than individuals with a lower percentage of *Wolbachia* reads (*Figure 3*, central panel: samples with black circles). Similarly, higher levels of *Wolbachia* in *Culex pipiens* mosquitoes showed a subtle shift in the number of distinct viral species detected (See *Figure 3—figure supplement 1* for a distinct view of the potential relationship between the number of Wolbachia reads and number of co-infecting viral species). Although not statistically significant, given the low sample numbers and lack of *Wolbachia* positive and negative individuals, these data again demonstrate the potential utility of sequencing individuals.

## Prevalence of eukaryotic microbes and pathogens in single mosquitoes

Although we detected fungi, plants and other eukaryotes in our analyses (*Figure 2*), we focus here on three potentially human pathogenic species: *Trypanosomatidae*, which was the most abundant eukaryotic taxon detected and contains established pathogens of both humans and birds; *Apicomplexa*, which encompasses the causative agents of human and avian malaria; and *Nematoda*, which contain filarial species that cause heartworm in canines and filarial diseases in humans.

Twelve mosquitoes (8%) were found to harbor *Trypanosomatidae* taxa (*Figure 3*, bottom panel). We detected sequences corresponding to monoxenous (e.g. *Crithidia* and species), dixenous (*Trypanosoma*, *Leishmania* species), as well as the more recently described *Paratrypanosoma confusum* species. Of the *Trypanosomatidae*-positive mosquitoes, eight were *Culex erythrothorax* mosquitoes, while the remaining four were *Culex pipiens* and two *Culex tarsalis Figure 3*, bottom panel. Notably, all were collected from the same trap site in Alameda County, albeit at different times, providing insight into, in this case, a limited distribution and potential prevalence of *Trypanosomatidae* within the mosquito population.

We investigated the distribution of the *Apicomplexa* contigs and reads, as this phylum encompasses the *Plasmodium* genus, which includes several pathogenic species that cause avian and human malaria. Within our single mosquito dataset, we identified eight mosquitoes with *Apicomplexa* contigs (*Figure 3*, bottom panel). These corresponded to three *Aedes aegypti* mosquitoes and one *Culex quinquefasciatus* mosquito, both collected in San Diego, and two *Culex erythrothorax* mosquitoes, one *Culex pipiens* mosquito, and one *Culex tarsalis* mosquito collected in Alameda County. Only the *Culex quinquefasciatus* mosquito harbored *Apicomplexa* reads at a level above 1% of total non-host reads. Interestingly, this mosquito also harbored *Wolbachia*, but no viruses could be detected.

Finally, we examined taxa falling under *Nematoda*, a phylum that encompasses a diverse set of more than 50 filarial parasites of humans and animals. Here, we saw evidence of *Nematoda* carriage in three *Culex* mosquitoes: two *Culex tarsalis* and one *Culex pipiens* (*Figure 3*, bottom panel). Two of these mosquitoes were collected in Alameda County and showed very low levels *Nematoda* (<1% of non-host reads, *Figure 3*, bottom panel, dark gray square symbols). In the third mosquito, a *Culex tarsalis* collected in Coachella Valley, the *Nematoda* made up 2% of the non-host reads (*Figure 3*, bottom panel gray bar with red outline).

Together these data reveal the diversity and prevalence of microbial species harbored within single mosquitoes and establish the comprehensive nature and sensitivity of single mosquito metagenomic analysis.

## Blood meals and associated microbes

As vectors, mosquitos transfer the pathogenic microbes they carry from one animal to another as they feed. Identifying the sources of these blood meals can provide critical information regarding the animal reservoir of these vector-pathogens and the paths of transmission. Therefore, we next investigated the possibility of identifying the blood meal host directly from mNGS. We restricted this analysis to the 60 mosquitoes from Alameda County, as they were selected for visible blood-engorgement. For 45 of the 60 mosquitoes, there was at least one contig with an LCA assignment to the phylum *Vertebrata* (range = 1–11 contigs, with 4–12,171 supporting reads). To assign a blood meal host for each of these mosquitoes, we compiled their corresponding *Vertebrata* contigs and selected the lowest taxonomic group consistent with those contigs. For all samples, the blood meal call fell into one of five broad categories (*Figure 5* and *Figure 5—source data 1*): even-toed ungulates (*Pecora*), birds (*Aves*), carnivores (*Carnivora*), rodents (*Rodentia*), and rabbits (*Leporidae*). For 10 samples, we were able to identify the genomic source at the species level, including rabbit (*Oryctolagus cuniculus*), mallard duck (*Anas platyrhynchos*), and raccoon (*Procyon lotor*).

The potential blood meal sources identified were broadly consistent with the habitats where the mosquitoes were collected. For the 25 samples collected in or near the marshlands of Coyote Hills Regional Park, we compare our calls to the wildlife observations in iNaturalist, a citizen science project for mapping and sharing observations of biodiversity. iNaturalist reports observations consistent with all five categories, including various species of squirrel, rabbit, raccoon, muskrat, and mule deer. The mosquitoes with blood meals in *Pecora* are likely feeding on mule deer, as no other ungulate commonly resides in that marsh (*iNaturalist, 2020*).

We also investigated whether bloodborne pathogens of the blood meal source were detectable. We performed a hypergeometric test for association between each blood meal category and each microbial taxon (see Materials and methods and *Figure 5—source data 2*). The only statistically significant association (p=0.0005, Bonferroni corrected) was between *Pecora* and *Anaplasma*, an intracellular erythroparasite transmitted by ticks. *Anaplasma* was detected in 11 of the 20 samples with *Pecora*. This striking co-occurrence suggests a possible burden of anaplasmosis in the local deer population. Additionally, we detected evidence for three other bloodborne pathogens which,

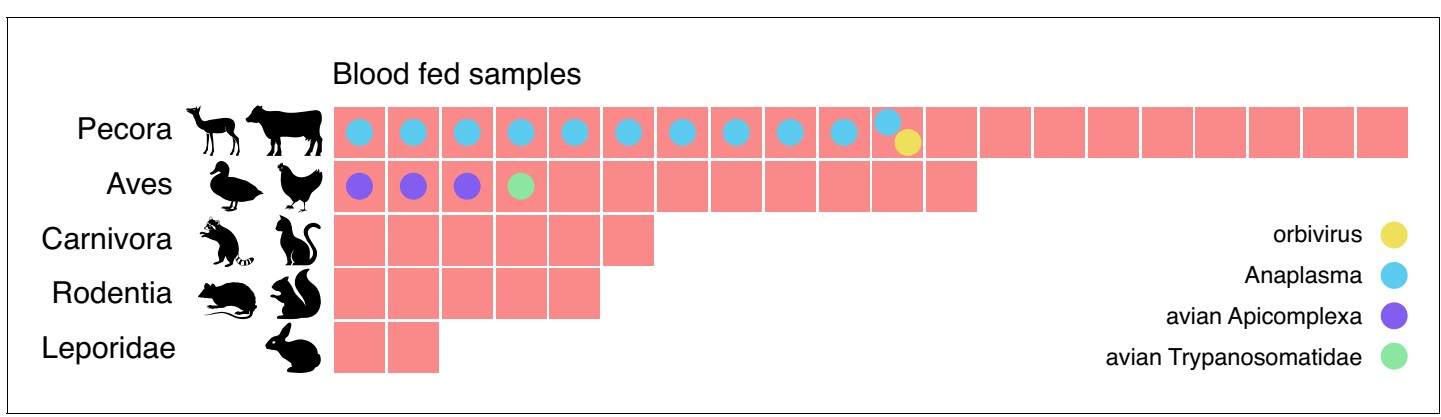

**Figure 5.** Metagenomic identification of sources of blood meals in individual mosquitos. Consensus taxonomic calls of vertebrate contigs for 45 of 60 blood fed mosquitoes collected in Alameda County. The remaining 15 samples had no vertebrate contigs. Red blocks represent individual mosquito samples; colored circles represent co-occurring contigs matching Orbivirus, Anaplasma, Avian Apicomplexa and Avian Trypanosomatidae representing possible bloodborne pathogens of the blood meal host.

The online version of this article includes the following source data for figure 5:

**Source data 1.** Data underlying *Figure 5* blood meal sources.
**Source data 2.** Data underlying *Figure 5* microbe calls.
**Source data 3.** Data underlying *Figure 5* virus and eukaryote calls.

because of the small number of observations, could not pass the threshold of statistical significance. These included an orbivirus closely related to those known to infect deer, a *Trypanosoma* species previously found in birds, and the apicomplexans *Plasmodium* and *Eimeria* from species known to infect birds (*Figure 5—source data 3*). The likely hosts of these pathogens were also concordant with the blood meal calls. Thus, sensitive and comprehensive metagenomic analysis of single mosquitoes not only provides information as to paths of transmission, it also provides a tool to detect emerging pathogens within animal communities in their environments.

## Recovery and assignment of previously unrecognizable viral genome segments and species within the orthomyxovirus family

Although many new viruses can be identified in bulk samples, the majority of these are identified only via their conserved RNA-dependent RNA polymerase (RdRp). Recovering complete genomes for segmented viruses from bulk samples is challenging, as genes that are not highly conserved may be unrecognizable by sequence homology. Moreover, the assignment of putative segments to a single genome can be confounded if the pooled libraries are derived from mosquitoes with multiple infections of related segmented viruses.

By sequencing many individual mosquitoes, we can exploit the fact that all segments of a segmented virus will co-occur in the samples where that virus is present and be absent in samples where the virus is absent. Applying these criteria to our data analysis should enable the identification of previously unidentified viral genome segments. To do this, we first grouped all contigs that were longer than 500 nucleotides into clusters of highly homologous contigs, then grouped these clusters by co-occurrence across all of the 148 individual mosquitos sampled (*Figure 6A*). Importantly, this required only the sequence information from the study, without using any external reference. We then scanned each cluster for sequences containing a viral RdRp domain (see Materials and methods). For each RdRp cluster, we consider any other contig cluster whose sample group overlaps the set of samples in the viral RdRp cluster above a threshold of 80% as a putative segment of the corresponding virus. A cluster-by-sample heatmap for all segments co-occurring with RdRps resulted in 27 candidate complete genomes for segmented viruses (*Figure 6—figure supplement 1*, *Figure 6—figure supplement 1—source data 1*, and *Figure 6—figure supplement 1— source data 2*). Out of a total of 145 contig clusters, 75 were non-RdRp segments. Of these 75 non-RdRp segments 60 bore recognizable homology (colored in black) to their expected counterparts, based on associated RdRp segments and 15 were linked to recognizable partial genomes via co-occurrence. This supports the notion that the remaining 15 putative segments (colored in red), which lack homology to any known sequences at either nucleotide or amino acid level, may indeed be part of viral genomes. Combined, these putative segments represented 7% of the metagenomic 'dark matter' portion of the reads in the study.

Our co-occurrence analysis enabled the discovery of new viral segments and new viral species within the segmented *Orthomyxoviridae* family (*Figure 6B* and *Figure 6—source data 1*). Orthomyxoviruses are segmented viruses (ranging from 6 to 8 segments) including influenza viruses, isaviruses, thogotoviruses, and quaranjaviruses that infect a range of vertebrate and arthropod species. Quaranjaviruses are largely found in arthropods, and in this study, we identified four quaranjaviruses, two of which were previously observed in mosquitoes collected outside California (Wuhan Mosquito Virus 6 [WMV6; *Li et al., 2015*; *Shi et al., 2017*] and Guadeloupe mosquito quaranja-like virus 1 [GMQV1; *Shi et al., 2019*]) and two, which we have named Ūsinis virus and Astopletus virus, were previously unknown.

Thus, for WMV6 and GMQV1 detected here, we observed all the previously identified segments (*Li et al., 2015*; *Shi et al., 2017*), as well as two additional segments (which we named hypothetical 2 and hypothetical 3) for WMV6 and five for GMQV1 (PA, gp64, hypothetical, hypothetical 2, hypothetical 3) (*Figure 6—figure supplement 2B*, panels A - D). We confirmed the existence of the two putative segments for WMV6 by assembling homologous segments from reads in two previously published datasets describing this virus. For GMQV1, we were able to find reads in NCBI's short read archive entries that are similar at the amino acid level to putative protein products of the two new segments; however, there was not sufficient coverage to reconstruct whole segments. Furthermore, phylogenetic trees constructed separately for each of the eight segments of WMV6 have similar topologies (See tanglegram, *Figure 7* and *Figure 7—source data 1*), suggesting that the two

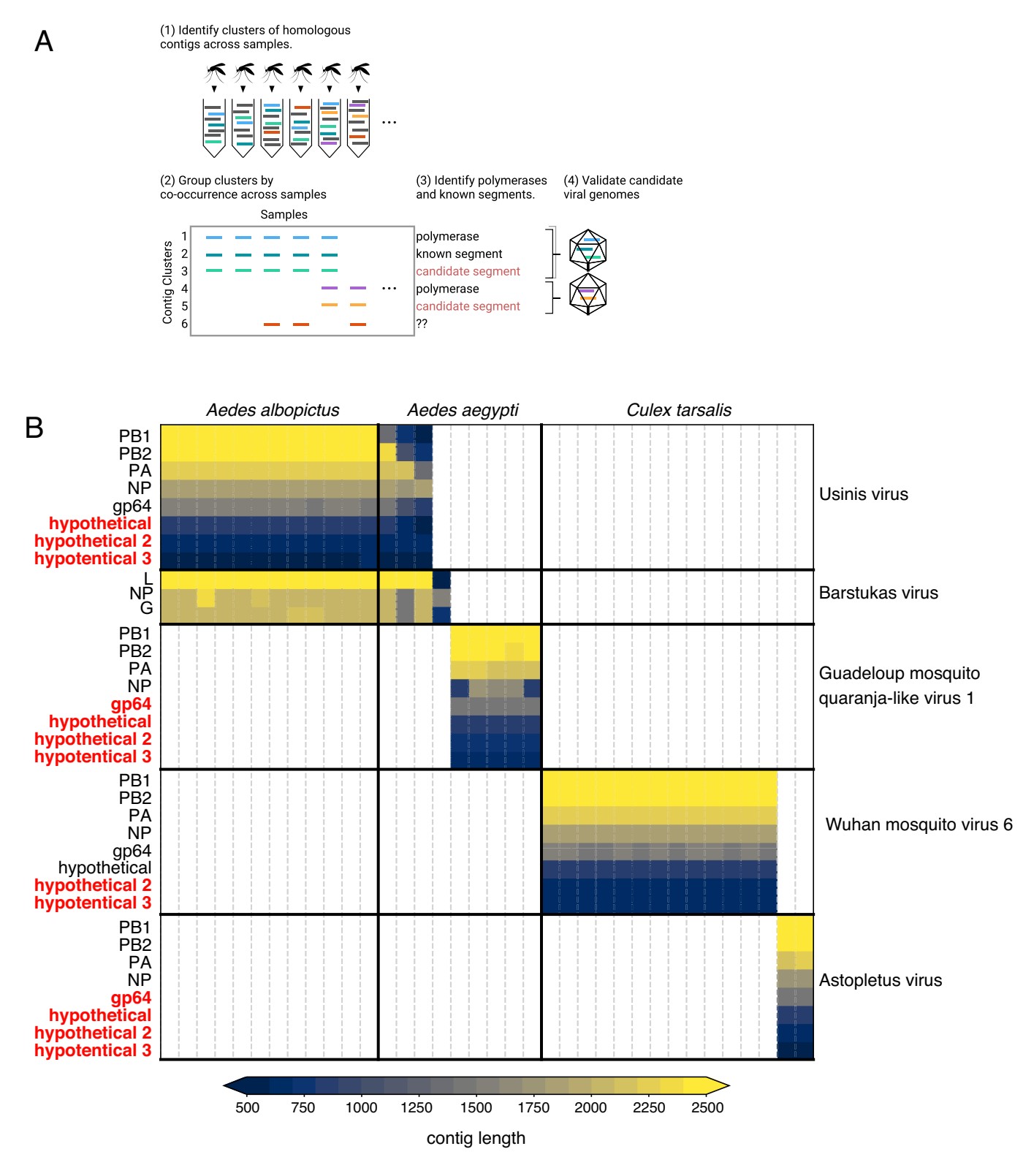

**Figure 6.** Bringing viral genomic dark matter into the light through single mosquito next generation metagenomic sequencing. (**A**) Previously unrecognized viral genome segments were identified among unaligned 'dark matter' contigs using co-occurrence analysis, which assumes that all segments of a segmented virus will co-occur in the samples where that virus is present and be absent in samples where the virus is absent. (**B**) Matrix of contigs derived from four distinct Orthomyxoviruses and one Phasma-like virus that were detected via their distinct co-occurrence pattern across

*Figure 6 continued on next page*

*Figure 6 continued*

mosquitoes. Rows are clusters of highly similar (99% identity) contigs and columns are individual mosquito samples. Light gray vertical lines delineate mosquito samples, dark black vertical lines indicate boundaries between mosquito species of each sample. Dark horizontal lines delineate segments comprising viral genomes. Labels on the right indicate viruses, with genomes delineated by horizontal lines. Guadeloupe mosquito quaranja-like virus one and Wuhan mosquito virus six were previously described and Ūsinis, Barstukas and Astopletus were named here. At left, plain text indicates putative labels for homologous clusters; black text indicates segments identifiable via homology (BLASTx) and red text indicates contig clusters that co-occur with identifiable segments but themselves have no identifiable homology to anything in GenBank. The Phasma-like Barstukas virus exhibits a nearly perfect overlap with Ūsinis virus (except for one sample in which Ūsinis was not found) but is identifiable as a Bunya-like virus due to having a three-segmented genome with recognizable homology across all segments to other Phasma-like viruses. Cells are colored by contig lengths (see color scale legend), highlighting their consistency which is expected of genuine segments. Deviations in detected contig lengths (e.g. *Aedes aegypti* samples that harbor shorter Ūsinis virus genome segments) reflect the presence of partial or fragmented contig assemblies in some of the samples.

The online version of this article includes the following source data and figure supplement(s) for figure 6:

**Source data 1.** Data underlying *Figure 6*, panel B.
**Figure supplement 1.** Identifying novel RNA segments in the 'dark matter' by co-occurrence.
**Figure supplement 1—source data 1.** Data underlying *Figure 6—figure supplement 1*.
**Figure supplement 1—source data 2.** Data underlying read fractions plotted in *Figure 6—figure supplement 1*.
**Figure supplement 2.** Co-occurrence in individual mosquitoes allows identification of novel RNA genome segments in mosquito-borne viruses.
**Figure supplement 3.** RdRp-based maximum likelihood tree spanning the quaranjaviruses in this study for which eight segments were recovered.
**Figure supplement 3—source data 1.** Data underlying *Figure 6—figure supplement 3*.
**Figure supplement 4.** Co-occurrence analysis enables identification of novel second narnavirus ambigrammatic RNA segment.
**Figure supplement 4—source data 1.** Data underlying *Figure 6—figure supplement 4*.
**Figure supplement 5.** Investigation of co-occurrences of narnaviruses and fungi in mosquitoes.
**Figure supplement 5—source data 1.** Data underlying *Figure 6—figure supplement 5*.

new putative segments have evolved in conjunction with the previous six, bringing the total number of segments for each genome to eight.

For the two quaranjaviruses discovered in this study, Ūsinis virus and Astopletus virus, the co-occurrence analysis also produced eight segments, 5 and 4 of which, respectively, were recognizable by alignment to NCBI reference sequences. The hypothetical 2 and hypothetical 3 segments we identified from this set of four quaranjavirus genomes are too diverged from one another to align via BLASTx, but they do share cardinal features such as sequence length, ORF length, and predicted transmembrane domains (*Figure 6—figure supplement 2*, panels A-D). Intriguingly, this set of four

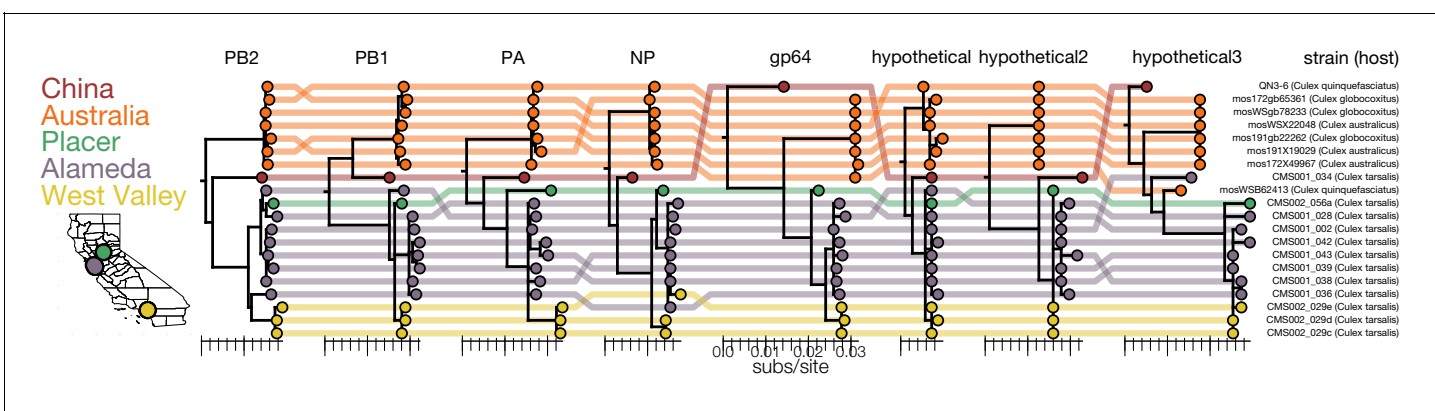

**Figure 7.** Genomic evidence for segment reassortment and intercontinental spread of mosquito-borne quaranjaviruses. (**A**) The Chinese strain (QN3-6) was originally described from a single PB1 sequence, while Australian (orange) viruses were described as having six segments. In this study we report the existence of two additional smaller segments (named hypothetical 2 and hypothetical 3) which we have assembled from our samples and the SRA entries of Chinese and Australian samples. Strains recovered in California as part of this study are colored by sampling location (Placer County in green, Alameda County in purple, West Valley in yellow). Strain names and hosts are indicated on the far right with colored lines tracing the position of each tip in other segment trees with crossings visually indicating potential reassortments.

The online version of this article includes the following source data for figure 7:

**Source data 1.** Data underlying *Figure 7*.

viruses are part of a larger clade of quaranjaviruses (*Figure 6—figure supplement 3* and *Figure 6—figure supplement 3—source data 1*). It is nearly certain that the remaining seven viruses in this clade also have eight segments and quite likely that all quaranjaviruses share this genome organization hinted at in earlier studies (*Zeller et al., 1989*).

The high rate of viral co-infections detected among the single mosquitoes we analyzed (*Figure 4*, panel A; *Figure 4—source data 1*) indicated a concomitant high likelihood that multiple mosquitoes could harbor more than one segmented virus, and potentially confound our co-occurrence analysis. However, the co-occurrence threshold of 0.8 that we applied was sufficient to deconvolve those segments into distinct genomes in all cases but one. There were 15 mosquito samples containing both Ūsinis virus, an orthomyxovirus with eight segments (three of which were unrecognizable by BLASTx) and Barstukas virus, a Phasma-like bunyavirus, with one additional sample where only Barstukas virus was found (*Figure 6B*, top two blocks). In this case, we were able to disentangle the genomes of these two viruses using additional genetic information: Barstukas virus contains all three segments expected for a bunyavirus (L, GP, and NP), all of which had BLASTx hits to other Phasma-like viruses, while the unrecognizable segments of Ūsinis virus shared features with the other quaranjaviruses in the study (as described above).

## Co-occurrence reveals unknown genome segments of Culex narnavirus 1

Beyond detection of missing genome segments for known segmented viruses, the co-occurrence analysis also revealed additional genome segments in 'dark matter' contig clusters for viruses with genomes previously considered to be non-segmented. A striking example is an 850 nucleotide contig cluster that co-occurred with the Culex narnavirus 1 RdRp segment in more than 40 mosquitoes collected from diverse locations across California (*Figure 6—figure supplement 1*, *Figure 6—figure supplement 1—source data 1*, and *Figure 6—figure supplement 1—source data 2*). Like the RdRp segment, the putative new second segment shares the exceptional feature of ambigrammatic open reading frames (ORFs), that is a distinct ORF encoded by the reverse complementary RNA strand (*Figure 6—figure supplement 2*, panel E). The phylogenetic tree topology for the set of 42 putative second segments is similar to the tree for the RdRp segments, suggesting co-inheritance (*Figure 6—figure supplement 4* and *Figure 6—figure supplement 4—source data 1*). Moreover, we were able to recover nearly identical contigs from previously published mosquito datasets, all of which also contained the Culex narnavirus 1 RdRp segment. This provides strong evidence that this otherwise unrecognizable sequence is a genuine Culex narnavirus one segment, which we refer to here as the 'Robin' segment, given it's consistent, but underappreciated presence.

Since the Narnaviruses were first described in fungi (*Hillman and Cai, 2013*) and recent studies have shown other eukaryotes can serve as Narnavirus hosts (*Charon et al., 2019*; *Dinan et al., 2019*; *Göertz et al., 2019*; *Richaud et al., 2019*), we investigated whether this virus co-occurred with a potential non-mosquito host. However, there was no significant co-occurrence with a non-mosquito eukaryotic taxon, or between the abundance of Culex narnavirus one and abundance of fungi (*Figure 6—figure supplement 5* and *Figure 6—figure supplement 5—source data 1*). Thus, it is likely that mosquitoes serve as direct hosts of the Culex narnavirus 1, whose genomes we show here consist of two, still enigmatic, ambigrammatic RNA segments.

## Validation of dark segments

In this study, we have described 14 novel *Orthomyxoviridae* and *Narnaviridae* segments. Some (Wuhan mosquito virus 6, Guadeloupe mosquito quaranja-like virus 1, and Culex narnavirus 1) were confirmed by assembly of novel segments from independent datasets, while other novel segments (of Ūsinis and Astopletus viruses) were confirmed by reference to apparent conserved features (similarity to other gp64 sequences, transmembrane domains, or splicing potential) and their descent from a common ancestor that most likely possessed these segments too. Only validation of the last dark segment we recovered, RNA 8 of Elemess virus, can thus be called cursory in comparison. As a cypo-like virus (*Reoviridae*), Elemess virus is expected to have at least ten segments. The co-occurrence of a random bit of RNA with consistent length across the same five samples where recognizable segments of Elemess virus were found is quite unlikely in its own right, and its lack of

recognizable homology to any non-viral species on GenBank or via HHpred, makes it more likely to be a genuine viral segment.

## Discussion

We demonstrate how mNGS of single mosquitoes, together with reference-free analyses and public databases, provides – in a single assay – critical and actionable epidemiological information. This includes quantitative information regarding circulating mosquito species, pathogen prevalence, and co-occurrence of diverse known and novel viruses, as well as prokaryotes, eukaryotes, blood meal sources and their potential pathogens. We are able to identify and confirm, using public data, the existence of previously unknown segments of both known and novel viruses, focusing on four quaranjaviruses and Culex narnavirus one as examples. In the context of an emerging disease, where knowledge about vectors, pathogens, and reservoirs is lacking, the techniques described here can be applied to rapidly provide actionable information for public health surveillance and intervention decisions. While unbiased sequencing of individual mosquitoes is not currently practical or appropriate in all contexts, advances in lab automation and rapidly decreasing costs of mNGS technologies are expected to increase the affordability and practicality of single mosquito sequencing in the near future.

### Inferring biology from sequence in the context of an incomplete reference

The power of metatranscriptomic NGS depends on the ability to extract biological information from nucleic acid sequences. For both bulk and single mosquito sequencing studies, the primary link between sequence and biology is provided by public reference databases, and thus the sensitivity of these approaches will depend crucially on the quality and comprehensiveness of those references. In practice, even the largest reference databases, such as nt/nr from NCBI, represent a small portion of the tree of life. Consequently, sequences derived from a sample of environmental or ecological origin, often exhibit only a low percent identity to even the best match in a database. Here, we manage that uncertainty by assigning a sequence to the lowest common ancestor of its best matches in the reference database. However, there is a fundamental limit to the precision of taxonomic identification from an incomplete reference.

An advantage of single mosquito sequencing is that it offers an orthogonal source of information: the ability to recognize nucleic acid sequences detected in many samples even when they have no homology to a reference sequence. This allowed us to associate unrecognizable sequences with viral polymerases, generating hypothetical complete genomes. The strategy of linking contigs that co-occur across samples is utilized in analysis of human and environmental microbiomes, where it is referred to as 'metagenomic binning' (*Breitwieser et al., 2019*; *Roumpeka et al., 2017*). Using this approach, we identified previously unknown genome segments establishing that the genome of a large clade of quaranjaviruses (those descended from the common ancestor of WMV6 and GMQV1), like distantly related influenza A and B viruses, consists of 8 segments. We also discovered a second ambigrammatic RNA encoded by the Culex narnavirus that in retrospect was identifiable in multiple previously published mosquito datasets. In sum, we pulled 7% of the reads in the metagenomic 'dark matter' fraction of our dataset into the light. The putative complete genomes we identified were supported retrospectively by public datasets and can be further validated by biological experiments or approaches such as short RNA sequencing that indicate a host antiviral response (*Aguiar et al., 2015*; *Waldron et al., 2018*).

Another advantage of single mosquito sequencing is the ability to supplement, or potentially circumvent, visual species identifications using molecular data. Accurate mosquito species identification is essential for the control of mosquito-borne diseases, as pathogen competence is often limited to a range of species, such as various *Aedes* species for Zika, dengue, and chikungunya viruses, and *Anopheline* species for malaria. Also, the primary mosquito species responsible for vectoring a disease can vary geographically – West Nile virus has been detected in 65 mosquito species, but a narrow range of *Culex* species drives transmission of the virus. Field validation of which mosquito species carry which pathogens in a specific geographic area informs targeted analysis and control of that species (*Petersen et al., 2013*). Here, though only 3 of the 10 collected mosquito species had complete genome references, it was possible to estimate pairwise SNP distances

between samples in a reference-free way and perform an unsupervised clustering. The clusters were 95% concordant with the visual mosquito species calls, and discordant outliers were easy to detect and correct. This approach generalizes to any collection of metatranscriptomes containing multiple representatives of each species. Accurate mosquito identification is essential for selecting the appropriate strategy and materials to control viremic mosquitoes. In a bulk pool of mosquitoes, the microbiota from any miscalled specimens would be blended in with the correctly labeled ones, making it difficult or impossible to deconvolute host species after the fact. By correctly identifying the host range of a known or novel pathogen in a given area, vector control can be appropriately targeted for the prevention of disease.

## Distribution of microbes within mosquito populations

Once sequences have been mapped to taxa, it is relatively straightforward to characterize the composition of the microbes within a circulating population of mosquitoes. This information can inform basic research and epidemiologic questions relevant for modeling the dynamics of infectious agents and the efficacy of interventions. A key parameter is the prevalence of a microbe, which cannot be inferred from bulk data. The 70 viruses identified in this study provide a compelling example. We found the prevalence of viruses ranged from detection in one mosquito (peribunya-like Udune virus) to detection in all 36 *Culex tarsalis* samples in the study (Marma virus).

A significant fraction (n = 46) of the 70 viral genomes we identified in this study correspond to novel divergent viruses. Further studies will be required to understand each of these viruses, and whether they correspond solely to microbial cargo (i.e. non-infecting viral species hitching a ride on the mosquito exterior or via ingestion of blood or nectar), versus insect specific viruses, or potentially transmissible human or animal pathogens. Among the 24 previously described viral genomes we recovered, we did not detect any of the known mosquito-borne human pathogens known to circulate in California (West Nile Virus, St. Louis Encephalitis Virus, Western Equine Encephalitis Virus). The lack of detection of these arboviruses likely reflects their low rate of circulation in the participating control districts during the sample collection period for the study. Indeed, both viruses were detected only sporadically and in only two of the five participating sites (see *Supplementary file 1* for an alignment of data from reports for mosquito pools testing positive for WNV and SLEV and sample collection statistics). In contrast, 10 of the 24 previously described viruses we recovered in this study correspond to viral agents described in a recent bulk mNGS of approximately 12,000 mosquitoes collected in California in 2016 (*Table 1*, *Sadeghi et al., 2018*) or an mNGS study of seven single mosquitoes collected in 2013 (*Chandler et al., 2015*). The remaining 15 previously described viral genomes have been observed outside of the state of California, in some cases on completely different continents. While technical differences in study design, types of mosquito populations examined, sample processing prior to RNA extraction, and sequencing approaches likely play a major role in the distinct set of viral genomes we report here, it is also likely that additional broader factors related to less well-understood aspects of the wild-caught mosquito virome and disease ecology – mainly the variability in the prevalence and distribution of mosquito viruses across locations, time, and mosquito host species – form the basis for these observations.

For some questions, the prevalence data supplied by single mosquito sequencing is helpful for experimental design. For example, in our dataset, *Wolbachia* was either absent or endemic in each mosquito species sampled. Thus, although a trend between the amount of *Wolbachia* relative to viral diversity was detectable across samples that harbored *Wolbachia*, it was not possible to detect a statistically significant effect of *Wolbachia* on virome composition or abundance within any species. Nonetheless, our data establish that single mosquito sequencing could address such questions via more extensive, targeted sampling of mosquito populations where *Wolbachia* (or any other agent of interest) is expected to have an intermediate prevalence. This information would be invaluable, as the introduction of *Wolbachia* might be a useful biological agent to suppress viral transmission by mosquitoes (*Moreira et al., 2009*).

## Blood meal sources and xenosurveillance

The identification of blood meal hosts is important for understanding mosquito ecology and controlling mosquito-borne diseases. Early field observations were supplemented by serology, and, more recently, molecular methods based on host DNA. Currently, the most common method of blood

meal identification is targeted PCR enrichment of a highly-conserved 'barcode' gene, such as mitochondrial cytochrome oxidase I, followed by sequencing (*Ratnasingham and Hebert, 2007*; *Reeves et al., 2018*). To monitor specific relationships between mosquito, blood meal, and pathogen, studies have combined visual identification of mosquitoes, DNA barcode identification of blood meal, and targeted PCR or serology for pathogen identification (*Batovska et al., 2018*; *Boothe et al., 2015*; *Tedrow et al., 2019*; *Tomazatos et al., 2019*).

Here, we extend the spectrum of molecular methods and show that unbiased mNGS of single mosquitoes can identify blood meal hosts, while simultaneously validating the mosquito species and providing an unbiased look at the pathogens. For these analyses, it is crucial that single mosquitoes were sequenced—if the mosquitoes had been pooled, it would not have been possible to associate potential vertebrate pathogens with a specific blood meal host. This allows for both reservoir identification, which seeks to identify the unknown host of a known pathogen, and xenosurveillance, which seeks to identify the unknown pathogens of specific vertebrate populations (*Grubaugh et al., 2015*). For example, in this study, we found a high prevalence of the tick-borne pathogen *Anaplasma* in mosquitoes that had likely ingested a blood meal from deer. Likewise in one of the deer-fed mosquitoes, we detected Lobuck virus, a novel orbivirus isolate that belongs to a clade of viruses implicated in a disease of commercially farmed deer reported in Missouri, Florida, and Pennsylvania (*Ahasan et al., 2019b*; *Ahasan et al., 2019a*; *Cooper et al., 2014*), but not California. This sort of novel virus-blood meal host observation provides a starting point to inform an understanding of the interplay between viruses and microbes circulating within the animal and insect populations in a given location. Ultimately, directed analyses in the lab and the field would be required to establish if potential virus-host observations like this Lobuck virus – deer example are (a) indeed present in the putative animal host populations, and (b) can be taken up via blood meal, and maintained and transmitted by mosquitoes.

Detection of a potential blood meal host in 45/60 of the blood engorged mosquitoes via RNA-based mNGS analysis is encouraging, especially in light of additional evidence of accompanying microbial cargo. The lack of success across all blood engorged mosquitoes may reflect factors that contribute to insufficient blood meal host RNA in the final extracted samples, such as variability in actual blood engorgement between mosquitoes, insufficient amounts of RNA circulating in the blood of certain types of animal hosts, or degradation/digestion of the blood meal RNA within the mosquito prior to extraction. Second, there may be too little information content in the RNA present, that is if it is from a highly conserved portion of a universal gene, there may not be an LCA below 'cellular organisms' or 'eukaryota'. A hybrid approach, in which primers for the enrichment of conserved, highly expressed 'barcode' genes described above are incorporated at the reverse transcription step (similar to metagenomic sequencing with spiked primer enrichment strategy (MSSPE) that has been applied to enrich specific viral species *Deng et al., 2020*) may provide a path forward to boost both the recovery rate and resolution of blood meal host identification in the context of RNA-based mNGS analyses.

## A critical role for public data in public health

This study would have been impossible without rich public datasets containing sequences, species, locations, and sampling dates. These provided the backbone of information allowing us to identify the majority of our sequences. Citizen scientist resources, such as the iNaturalist catalog of biodiversity observations, was a valuable complement, providing empirical knowledge of species distributions in the mosquito collection area that resolved the ambiguity we detected in sequence space.

In sum, complementing conventional analyses of mosquito pools and field observations of mosquitoes and the animals they bite with single mosquito mNGS can provide valuable complementary information to enhance the evidence base for distinct interventions to control mosquito-borne infectious diseases. As shown here, single mosquito mNGS can map an uncharted landscape related to the movement of pathogens between mosquitoes and their reservoirs. This can inform the deployment of targeted detection or surveillance assays for both established and emerging mosquito-borne pathogens across large geographical areas or animal reservoir populations. As mosquitoes and their microbiota continue to evolve and migrate, posing new risks for human and animal populations, these complementary approaches will empower scientists and public health professionals.

## Materials and methods

### Mosquito collection

The 148 adult mosquitoes included in this study were collected at sites indicated in *Figure 1—source data 1* using encephalitis virus survey (EVS) or gravid traps that were baited with $CO_2$ or hay-infused water, respectively. The collected mosquitoes were frozen using dry ice or paralyzed using triethyl amine and placed on a −15°C chill table or in a glass dish, respectively, for identification to species using a dissection microscope. Identified female mosquitoes were immediately frozen using dry ice in deep well 96-well plates and stored at −80°C or on dry ice until the nucleic acids were extracted for sequencing.

### RNA preparation

Individual mosquitoes were homogenized in bashing tubes with 200uL DNA/RNA Shield (Zymo Research Corp., Irvine, CA, USA) using a 5mm stainless steel bead and a TissueLyserII (Qiagen, Valencia, CA, USA) (2x1 min, rest on ice in between). Homogenates were centrifuged at 10,000xg for 5 min at 4°C, supernatants were removed and further centrifuged at 16,000xg for 2 min at 4°C after which the supernatants were completely exhausted in the nucleic acid extraction process. RNA and DNA were extracted from the mosquito supernatants using the ZR-DuetTM DNA/RNA MiniPrep kit (Zymo Research Corp., Irvine, CA, USA) with a scaled down version of the manufacturer's protocol with Dnase treatment of RNA using either the kit's DNase or the Qiagen RNase-Free DNase Set (Qiagen, Valencia, CA, USA). Water controls were performed with each extraction batch. Quantitation and quality assessment of RNA was done by the Invitrogen Qubit 3.0 Fluorometer using the Qubit RNA HS Assay Kit (ThermoFisher Scientific, Carlsbad, CA, USA) and the Agilent 2100 BioAnalyzer with the RNA 6000 Pico Kit (Agilent Technologies, Santa Clara, CA, USA).

### Library prep and sequencing

Up to 200 ng of RNA per mosquito, or 4 µL aliquots of water controls extracted in parallel with mosquitoes, were used as input into the library preparation. A 25 pg aliquot of External RNA Controls Consortium (ERCC) RNA Spike-In Mix (Ambion, ThermoFisher Scientific, Carlsbad, CA, USA) was added to each sample. The NEBNext Directional RNA Library Prep Kit (Purified mRNA or rRNA Depleted RNA protocol; New England BioLabs, Beverly, MA, USA) and TruSeq Index PCR Primer barcodes (Illumina, San Diego, CA, USA) were used to prepare and index each individual library. The quality and quantity of resulting individual and pooled mNGS libraries were assessed via electrophoresis with the High Sensitivity NGS Fragment Analysis Kit on a Fragment Analyzer (Advanced Analytical Technologies, Inc), the High-Sensitivity DNA Kit on the Agilent Bioanalyzer (Agilent Technologies, Santa Clara, CA, USA), and via real-time quantitative polymerase chain reaction (qPCR) with the KAPA Library Quantification Kit (Kapa Biosystems, Wilmington, MA, USA). Final library pools were spiked with a non-indexed PhiX control library (Illumina, San Diego, CA, USA). Pair-end sequencing (2 × 150 bp) was performed using an Illumina NovaSeq or NextSeq sequencing system (Illumina, San Diego, CA, USA). The pipeline used to separate the sequencing output into 150-base-pair pair-end read FASTQ files by library and to load files onto an Amazon Web Service (AWS) S3 bucket is available on GitHub at https://github.com/czbiohub/utilities.

### Mosquito species validation

To validate and correct the visual assignment of mosquito species, we estimated SNP distances between each pair of mosquito transcriptomes by applying SKA (Split Kmer Analysis) (*Harris, 2018*) to the raw fastq files for each sample. The hierarchical clustering of samples based on the resulting distances was largely consistent with the visual assignments, with each cluster containing a majority of a single species. To correct likely errors in the visual assignment, samples were reassigned to the majority species in their cluster, resulting in 7 changes out of 148 samples and one species assignment for a sample lacking a visual assignment.

## Host and quality filtering

Raw sequencing reads were host- and quality-filtered and assembled using the IDseq (v3.2) (*Kalantar et al., 2020*) platform https://idseq.net, a cloud-based, open-source bioinformatics platform designed for detection of microbes from metagenomic data.

## Host reference

We compiled a custom mosquito host reference database made up of:

1. All available mosquito genome assemblies under NCBI taxid 7157 (*Culicidae*; n = 41 records corresponding to 28 unique mosquito species, including 1 *Culex*, 2 *Aedes*, and 25 *Anopheles* records) from NCBI Genome Assemblies (accession date: 12/7/2018).
2. All mosquito mitochondrial genome records under NCBI taxid 7157 available in NCBI Genomes (accession date: 12/7/2018; n = 65 records).
3. A *Drosophila melanogaster* genome (GenBank GCF_000001215.4; accession date: 12/7/2018).

Mosquito Genome Assembly and mitochondrial genome accession numbers and descriptions are detailed in *Figure 2—source data 1*.

## Read filtering

To select reads for assembly, we used the IDseq platform to perform a series of filtering steps described below. Yields at each step for each sample are provided in *Figure 1—source data 2*. A detailed description of all parameters is available in the IDseq documentation (https://github.com/chanzuckerberg/idseq-dag/wiki/IDseq-Pipeline-Stage-%231:-Host-Filtering-and-QC).

### Filter host 1

Remove reads that align to the host reference using the Spliced Transcripts Alignment to a Reference (STAR) algorithm.

### Trim adapters

Trim Illumina adapters using trimmomatic.

### Quality filter

Remove low-quality reads using PriceSeqFilter.

### Remove duplicate reads

Remove duplicate reads using CD-HIT-DUP.

### Low-complexity filter

Remove low-complexity reads using the LZW-compression filter.

### Filter host 2

Remove further reads that align to the host reference using Bowtie2, with flag very-sensitive-local. The remaining reads are labeled 'non-host' in *Figure 1—source data 2*; Below we refer to these reads as 'putative non-host reads.'.

## Assembly

The putative non-host reads for each sample were assembled into contigs using SPADES (*Bankevich et al., 2012*) with default settings. The reads used for assembly were mapped back to the contigs using Bowtie2 (*Langmead and Salzberg, 2012*) (flag very-sensitive), and contigs with more than two reads were retained for further analysis.

## Taxonomic assignment

We aligned each contig to the nt and nr databases using BLASTn (*Altschul et al., 1990*) (discontinuous megablast) and PLAST (a faster implementation of the BLASTx algorithm), respectively. (The databases were downloaded from NCBI on Mar 27, 2019.) Default parameters were used, except

the E-value cutoff was set to 1e-2. For each contig, the results from the database (either nt or nr) with a better top hit (as judged by bitscore) were used for further analysis.

For contigs with BLAST hits to more than one species, we report the lowest common ancestor (LCA) of all hits whose number of matching aligned bases alignment length*percent identity is no less than the number of aligned bases for the best BLAST hit minus the number of mismatches in the best hit. (In the case that the same segment of the query is aligned for all hits, this condition guarantees that the excluded hits are further from the best hit than the query is.)

For 172,244 contigs, there were strong BLAST hits to *Hexapoda*, the subphylum of arthropods containing mosquitoes. This is likely a consequence of the limited number and quality of genomes used in host filtering, and all contigs with an alignment to *Hexapoda* of at least 80% of the query length or whose top hit (by e-value) was to *Hexapoda* were discarded from further analysis.

Contigs with no BLAST hits are referred to as 'dark contigs'.

For RNA viruses, where complete or near-complete genomes were recovered, a more sensitive analysis was performed (see below).

## Contamination and further host filtering

There are many potential sources of contaminating nucleic acid in mNGS analyses, including lab surfaces, human experimenters, and reagent kits. We attempt to quantify and correct for this contamination using eight water controls. Here, we model contamination as a random process, where the mass of a contaminant taxon $t$ in any sample (water or Mosquito) is a random variable $X_t$. We convert from units of reads to units of mass using the number of ERCC reads for each sample (as a fixed volume of ERCC spike-in solution was added to each sample well). We estimate the mean of $X_t$ using the water controls. We say that a taxon observed in a sample is a possible contaminant if the estimated mass of that taxon in that sample is less than 100 times the average estimated mass of that taxon in the water samples. Since the probability that a non-negative random variable is greater than 100 times its mean is at most 1% (Markov's inequality), this gives a false discovery rate of 1%. For each possible contaminant taxon in a sample, all contigs (and reads) assigned to that taxon in that sample were excluded from further analysis. A total of 46,603 reads were removed as possible contamination using this scheme. (Human and mouse were identified as the most abundant contaminant species.)

For every sample, 'non-host reads assembled into contigs' refers to reads mapping to contigs that pass the above filtering, *Hexapoda* exclusion, and decontamination steps. The generic term, 'non-host reads' encompasses these reads plus any other reads passing the the above filtering, *Hexapoda* exclusion, and decontamination steps, and failed to assemble into contigs or assembled into a contig with only two reads.

## Viral polymerase detection and segment assignment

Alignments of viral RNA-dependent RNA polymerases used to detect domains were downloaded from Pfam. These were RdRP_1 (PF00680, *Picornavirales*-like and *Nidovirales*-like), RdRP_2 (PF00978, *Tymovirales*-like and Hepe-Virga-like), RdRP_3 (PF00998, *Tombusviridae*-like and *Nodaviridae*-like), RdRP_4 (PF02123, *Toti*-, *Luteo*-, and *Sobemoviridae*-like), RdRP_5 (PF07925, *Reoviridae*-like), Birna_RdRp (PF04197, *Birnaviridae*-like), Flavi_NS5 (PF00972, *Flaviviridae*-like), Mitovir_RNA_-pol (PF05919, *Narnaviridae*-like), Bunya_RdRp (PF04196, *Bunyavirales*-like), Arena_RNA_pol (PF06317, *Arenaviridae*-like), Mononeg_RNA_pol (PF00946, *Mononega*- and *Chuviridae*-like), Flu_PB1 (PF00602, *Orthomyxoviridae*-like). Hidden Markov model (HMM) profiles were generated from these with HMMER (v3.1b2; http://hmmer.org/) and tested against a set of diverged viruses, including ones thought to represent new families. Based on these results only the RdRP_5 HMM was unable to detect diverged Reovirus RdRp, such as Chiqui virus. An additional alternative Reovirus HMM (*HMMbuild* command) was generated by using BLASTp hits to Chiqui virus, largely to genera *Cypovirus* and *Oryzavirus*, aligned with MAFFT (*Katoh et al., 2005*) (E-INS-i, BLOSUM30).

All contigs of length >500 base pairs were grouped into clusters using a threshold of $\geq$ 99% identity (CD-HIT-EST *Li and Godzik, 2006*). Representative contigs from each cluster were scanned for open reading frames (standard genetic code) coding for proteins at least 200 amino acids long, in all six frames with a Python script using Biopython (*Cock et al., 2009*). These proteins were scanned using HMM profiles built earlier and potential RdRp-bearing contigs were marked for follow up. We

chose to classify our contigs by focusing on RdRp under the assumption that *bona fide* exogenous viruses should at the very least carry an RdRp and be mostly coding-complete. Contigs that were not associated with an RdRp or coding-complete included Cell fusing agent virus (*Flaviviridae*, heavily fragmented) and Phasma-like nucleoprotein sequences (potential piRNAs) in a few samples.

## Co-occurrence

For each cluster whose representative contig contained a potential RdRp, we identified its putative viral segment from CD-HIT clusters whose set of samples overlapped the set of samples in the RdRp cluster at a threshold of 80%. (That is, a putative segment should be present in at least 80% of the samples that RdRp is present in, and RdRp should be present in at least 80% of the samples that the putative segment is present in).

In cases where a singleton segmented (bunya-, orthomyxo-, reo-, chryso-like, *etc*) virus was detected in a sample we relied on the presence of BLASTx hits of other segments to related viruses (e.g. diverged orthobunyavirus). We thus linked large numbers of viral or likely viral contigs to RdRps representing putative genomes for these lineages.

## Final classification

There were 1269 contigs identified as viral either by RdRp detection or co-occurrence, and the resulting species-level calls are used for further analysis in lieu of the LCA computed via BLAST alignments. This included 338 'dark contigs' which had no BLAST hits, 748 with LCA in Viruses; the LCAs for the remainder were Bacteria (9), and Eukaryota (4), and Ambiguous (170), a category including (including root, cellular organisms, and synthetic constructs). Reads are assigned the taxonomic group of the contig they map to.

Completeness of viral genomes was assessed by reference to known gene repertoires of virus groups (e.g. *Chrysoviridae* have four segments with four genes, *Dicistroviridae* have a monolithic genome and two genes, *etc*). A genomes was deemed complete if all genes expected in a particular virus group were present and partial if ORFs were terminated by shorter contigs. While the number of expected segments are known and seemingly conserved for many segmented virus groups and thus our inference for segmented genomes being complete should be correct, we also clearly showed that groups not often receiving much attention, like *Orthomyxoviridae*, *Narnaviridae* and *Reoviridae* can still prove surprising.

## Treemap

Treemaps (e.g. *Figure 2*) are a way of visualizing hierarchical information as nested rectangles whose area represents numerical values. To visualize the distribution of reads amongst taxonomic ranks, we first split the data into two categories: viral and cellular. For cellular taxonomic ranks (Bacteria, Eukaryotes, Archaea and their descendants) we assigned all reads of a contig to the taxonomic compartment the contig was assigned (see above, 'Taxonomic Assignment'). For viral taxa, we relied on the curated set of viral contigs coding for RdRp and their putative segments, where a putative taxonomic rank (usually family level) had been assigned. All the non-host reads assembled into contigs that comprised putative genomes were assigned to their own compartment in the treemap, under the curated rank. Additional compartments were introduced to either reflect aspects of the outdated and potentially non-monophyletic taxonomy which is nevertheless informative (e.g. positive- or double-strandedness of RNA viruses) or represent previously reported groups without an official taxonomic ID on public databases (e.g. Narna-Levi, Toti-Chryso, Hepe-Virga, *etc*).

To prototype the cellular part of the treemap, all taxonomic IDs encountered along the path from the assigned taxonomic ID up to root (i.e. the taxonomic ID's lineage) were added to the treemap. Based on concentrations of reads in particular parts of the resulting taxonomic treemap, prior beliefs about the specificity of BLAST hits, and information utility, this was narrowed down to the following taxonomic ranks: *Bacteria*, *Wolbachia*, *Gammaproteobacteria*, Spirochaetes, *Terrabacteria* group, Fungi, *Boroeutheria*, *Aves*, *Trypanosomatidae*, *Leishmaniinae*, *Viridiplantae*.

## Microbiota distribution in single mosquitoes

In *Figure 3*, the denominators are non-host reads. The numerators are numbers of reads from contigs with confident assignments. For viruses, these contigs came from viral curation or co-occurrence.

For *Wolbachia* and eukaryotes, these contigs had LCA assignment within the *Wolbachieae* tribe (taxid: 952) and *Eukaryota* superkingdom (taxid: 2759), respectively, and had a BLAST alignment where the percentage of aligned bases was at least 90%. Groups within viruses, *Wolbachia*, and eukaryotes were excluded for a given sample if the cumulative proportion of non-host reads was less than 1%. Samples were excluded if the total proportions of non-host reads belonging to viruses, *Wolbachia*, or eukaryotes were all below 1%.

## Blood meal calling

For each of the 60 blood fed mosquito samples from Alameda County, we selected each contig with LCA in the subphylum *Vertebrata*, excluding those contained in the order *Primates* (because of the possibility of contamination with human DNA). For each sample, we identified the lowest rank taxonomic group compatible with the LCAs of the selected contigs. (A taxonomic group is compatible with a set of taxonomic groups if it is an ancestor or descendent of each group in the set.) For 44 of the 45 samples containing vertebrate contigs, this rank is at class or below; for 12 samples, it is at the species level. Each taxonomic assignment falls into one of the following categories: *Pecora*, *Aves*, *Carnivora*, *Rodentia*, *Leporidae*. In **Figure 5**, each sample with a blood meal detected is displayed according to which of those categories it belongs (Underlying data for **Figure 5** are provided in Supplemental Data file bloodmeal_calls.tsv, with microbe categories tested for each sample summarized in samples_taxa.csv). The remaining sample, CMS001_022_Ra_S6, contained three contigs mapping to members of *Pecora* and a single contig with LCA *Euarchontoglires*, a superorder of mammals including primates and rodents; we annotate this sample as containing *Pecora*.

Notably, 19 samples contain at least one contig with LCA in genus *Odocoileus* and another contig with LCA genus *Bos*. While the lowest rank compatible taxonomic group is the infraorder *Pecora*, it is likely that a single species endemic in the sampled area is responsible for all of these sequences. Given the observational data in the region (described in the main text), that species is likely a member of *Odocoileus* whose genome diverges somewhat from the reference.

## Phylogenetic analyses

We chose a single Wuhan mosquito virus six genome from our study (CMS001_038_Ra_S22) as a reference to assemble by alignment the rest of the genome of strain QN3-6 (from SRA entry SRX833542 as only PB1 was available for this strain) and the two small segments discovered here for Australian segments (from SRA entries SRX2901194, SRX2901185, SRX2901192, SRX2901195, SRX2901187, SRX2901189, and SRX2901190) using Magic-BLAST (**Boratyn et al., 2018**). Due to much higher coverage in Australian samples, Magic-BLAST detected potential RNA splice sites for the smallest segment (hypothetical 3) which would extend the relatively short open reading frame to encompass most of the segment. Sequences of each segment were aligned with MAFFT (Auto setting) and trimmed to coding regions. For the hypothetical 3 segment, we inserted Ns into the sequence near the RNA splice site to bring the rest of the segment sequence into frame.

PhyML (**Guindon et al., 2010**) was used to generate maximum likelihood phylogenies under an HKY+$\Gamma_4$ model (**Guindon et al., 2010**; **Hasegawa et al., 1985**; **Yang, 1994**). Each tree was rooted via a least-squares regression of tip dates against divergence from root in TreeTime (**Sagulenko et al., 2018**). Branches with length 0.0 in each tree (arbitrarily resolved polytomies) were collapsed, and trees untangled and visualized using baltic21 (https://github.com/evogytis/baltic).

To generate the Culex narnavirus 1 tanglegrams, 42 sequences of RdRp and 42 co-occurring Robin segment sequences from our samples and three previously published RdRp sequences (MK628543, KP642119, KP642120) as well as their three corresponding Robin segments assembled from SRA entries (SRR8668667, SRR1706006, SRR1705824, respectively) were aligned with MAFFT and trimmed to just the most conserved open reading frame (as opposed to its complement on the reverse strand). Maximum likelihood phylogenies for both RdRp and Robin segments were generated with PhyML with 100 bootstrap replicates under an HKY+$\Gamma_4$ substitution model. The resulting phylogenies were mid-point rooted, untangled and visualized using balti (https://github.com/evogytis/baltic).

## Data and code availability

Raw and assembled sequencing data are deposited in NCBI Bioproject PRJNA605178. Code is available on Github at https://github.com/czbiohub/california-mosquito-study (copy archived at swh:1:rev:f09dd7835aa4abbe2fb96f7da73ed60dcff5b8b4; *Batson, 2021*). Derived data (including all contigs) and supplementary data are available on Figshare at http://dx.doi.org/10.6084/m9.figshare.11832999.

## Acknowledgements

We thank our collaborating partners in the California Mosquito and Vector Control Agency Districts of Alameda, Placer, San Diego, West Valley, and Coachella Valley, who provided all the mosquito specimens and corresponding metadata that made this study possible. We thank Maira Phelps for liaison work with collaborators and in-house specimen management. We thank Rene Sit, Michelle Tan, and Norma Neff of the Chan Zuckerberg Biohub Genomics Platform for supporting all aspects of mNGS sequencing for this study. We thank the IDseq team at the Chan Zuckerberg Initiative for useful discussions and facilitation of analysis over the course of this study. We thank Jack Kamm, Darren J Obbard, and Cristina Tato for useful discussions during the development of this project. We would also like to acknowledge Natalie Whitis and Annie Lo for their contribution to the early phases of the specimen extraction and sequencing library preparation for this project. We thank Sandra Schmid, Bill Burkholder, Cristina Tato, Peter Kim, David Yllanes, and Joe DeRisi for reviewing the manuscript.

## Additional information

### Funding

| Funder | Author |
| --- | --- |
| Chan Zuckerberg Initiative | Joshua Batson |
| University of California, San Francisco | Hanna Retallack |
| National Science Foundation | Kalani Ratnasiri |

The funders had no role in study design, data collection and interpretation, or the decision to submit the work for publication.

### Author contributions

Joshua Batson, Conceptualization, Data curation, Formal analysis, Supervision, Validation, Investigation, Visualization, Writing - original draft, Writing - review and editing; Gytis Dudas, Conceptualization, Data curation, Software, Formal analysis, Validation, Investigation, Visualization, Methodology, Writing - original draft, Writing - review and editing; Eric Haas-Stapleton, Conceptualization, Resources, Data curation, Investigation, Methodology, Writing - original draft, Project administration, Writing - review and editing; Amy L Kistler, Conceptualization, Resources, Data curation, Supervision, Validation, Investigation, Visualization, Methodology, Writing - original draft, Project administration, Writing - review and editing; Lucy M Li, Conceptualization, Data curation, Software, Formal analysis, Validation, Visualization, Methodology, Writing - original draft, Writing - review and editing; Phoenix Logan, Software, Formal analysis, Visualization, Methodology, Writing - review and editing; Kalani Ratnasiri, Data curation, Formal analysis, Investigation, Methodology, Writing - original draft, Writing - review and editing, Performed all experimental aspects of the study; Hanna Retallack, Conceptualization, Data curation, Software, Formal analysis, Investigation, Visualization, Methodology, Writing - original draft, Writing - review and editing

### Author ORCIDs

Gytis Dudas  http://orcid.org/0000-0002-0227-4158
Amy L Kistler  https://orcid.org/0000-0003-1112-719X
Lucy M Li  http://orcid.org/0000-0002-6562-4004

Kalani Ratnasiri (ID) http://orcid.org/0000-0001-5953-0004
Hanna Retallack (ID) http://orcid.org/0000-0003-0533-9102

**Decision letter and Author response**
Decision letter https://doi.org/10.7554/eLife.68353.sa1
Author response https://doi.org/10.7554/eLife.68353.sa2

## Additional files

### Supplementary files
• Supplementary file 1. Summary of West Nile Virus (WNV) and St. Louis encephalitis virus (SLEV) surveillance during the mosquito sample collection period.

• Transparent reporting form

### Data availability

Raw and assembled sequencing data are deposited in NCBI Bioproject PRJNA605178. Code is available on Github at https://github.com/czbiohub/california-mosquito-study (copy archived at https://archive.softwareheritage.org/swh:1:rev:f09dd7835aa4abbe2fb96f7da73ed60dcff5b8b4). Derived data (including all contigs) and supplementary data are available on Figshare at https://doi.org/10.6084/m9.figshare.11832999.

The following datasets were generated:

| Author(s) | Year | Dataset title | Dataset URL | Database and Identifier |
|---|---|---|---|---|
| Batson J, Dudas G, Haas-Stapleton E, Kistler AL, Li LM, Logan P, Ratnasari K, Retallack H | 2020 | Single mosquito metatranscriptomic sequence analysis to understand the full complement of microbial cargo harbored by individual mosquitoes and the animals they feed on. | https://www.ncbi.nlm.nih.gov/bioproject/PRJNA605178 | NCBI BioProject, PRJNA605178 |
| Batson J, Dudas G, Haas-Stapleton E, Kistler AL, Li LM, Logan P, Ratnasari K, Retallack H | 2021 | Metatranscriptomic sequencing of single mosquitos in California | https://doi.org/10.6084/m9.figshare.11832999 | figshare, 10.6084/m9.figshare.11832999 |

The following previously published datasets were used:

| Author(s) | Year | Dataset title | Dataset URL | Database and Identifier |
|---|---|---|---|---|
| Li C-X, Shi M, Tian J-H, Lin X-D, Kang Y-J, Chen L-J, Quin X-C, Xu J, Holmes EC, Zhang Y-Z | 2015 | Genome survey of RNA viruses within arthropods | https://www.ncbi.nlm.nih.gov/bioproject/PRJNA271540 | NCBI BioProject, PRJNA271540 |
| Shi M, Neville P, Nicholson J, Ede JS, Imrie A, Holmes EC | 2017 | Mosquitoes (mosquitoes raw sequence reads) | https://www.ncbi.nlm.nih.gov/bioproject/PRJNA388696 | NCBI BioProject, PRJNA388696 |

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
