## [Decision Letter]

[Editors' note: this paper was reviewed by Review Commons.]

**Acceptance summary:**

In this study, the authors utilized unbiased meta-transcriptomic in sequencing 148 diverse wild-caught mosquitoes (Aedes, Culex, and Culiseta mosquito species) collected in California. They generated an overwhelming dataset and thoughtful analysis on the vector species, the source of the blood meals and the microbiome/virome using a simple experimental approach and using single mosquitoes. They make a strong case for the power of mNGS of mosquitoes that may be applicable to other (invertebrate) species.

---

## [Author Response]

Reviewer #1 (Evidence, reproducibility and clarity (Required)):This is a very interesting and well-designed study on mNGS of mosquitoes. The authors demonstrate that they can distill valuable information on the vector species, the source of the blood meals and the microbiome/virome using a simple experimental approach and using single mosquitoes. A highlight of the work is that the paper is very comprehensive with an overwhelming dataset and thoughtful analysis. It is a showcase how sequencing data from a relative compact number of mosquitoes specimens can be used to conduct sophisticated computational analysis leading to meaningful conclusions. The authors make a strong case for the power of mNGS of mosquitoes that may be applicable to other (invertebrate) species. Especially the phylogenetic analysis based on SNP distance without have reference genomes and the grouping of contigs by means of co-occurence in datasets is original. We feel that the work deserves to be published.Reviewer #1 (Significance (Required)):We have a number of comments that the authors may consider in further improving the quality of their manuscript:What is the impact of this paper?I think it is possible that the paper will have a decent impact on the mosquito arbovirus field, because it adequately shows the possibilities that individual mosquito sequencing can bring (e.g. co-occurrence analysis). It may shift the balance to doing more individual mosquito sequencing instead of pools. The paper is also very extensive in the analyses that it does on this very rich data set. Below, some suggestions are given for additional analysis, which should be interpreted as a compliment to the interesting data set acquired. It should however be noted that the ideas and approaches taken are not entirely new. Sequencing individual mosquitoes, co-occurrence analysis and metagenomic sequencing have been done before, although not to this extent and not in this field. Several novel possibilities:1. An unbiased way to check if you have the correct mosquito species and the ability to detect subspecies. Using the genetic distance of the transcriptomes they have likely corrected the missed identification in some samples, where these calls had a logical mistake made. The fact that subspecies overlapped with the sites of capture is very interesting and confirms the relevance of looking at the genetic distance also within species.2. Blood-meal analysis from sequence data. Here they can get to species level for 10 out of 40 blood-engorged mosquitoes. The idea is interesting, as you would be able to get a lot more information if you can determine blood-meal origin from RNA-seq data (as shown in this paper).However, I feel that in the current paper (and this may be intentional) they do not properly show that RNA-seq is an adequate alternative to DNA sequencing of the blood. To convince me, I would have liked to have these results compared to DNA sequencing and see how much overlap there is. I understand however that the choice was made not to do this, but I do have a small note for the information given now.

Reviewer 1 correctly infers that our intention for this manuscript was not to demonstrate that RNA mNGS is a comparable or superior alternative to DNA sequencing. Rather, our aim was to perform an exploratory investigation to determine if insights into blood meal sources could actually be detected via RNA-based mNGS analysis, as we stated in the second sentence of Blood meals and associated microbes: “Therefore, we next investigated the possibility of identifying the blood meal host directly from mNGS.”

It was mentioned that 1 contig with an LCA of vertebrates is enough for a 'blood-meal origin' call. I am however left to wonder how reliable is 1 read?

We did not pursue analysis of single reads for the blood meal analysis. In our study, we used short read sequencing technology whereby a single read provides only a limited amount of sequence information (146bp). This presents a high probability of significant ambiguity in taxonomic assignment. Focusing on contigs as the unit of analysis, rather than reads, provided additional input sequence lengths for the blastn alignment steps of our analyses that we use for taxonomic assignment. Having longer input sequences at this step increases the probability of an informative and statistically significant blastn alignment result and associated taxonomic assignment.

Overall, for a sample to be included in the blood meal analysis, we required a minimum of 1 contig with a *Vertebrata* LCA assignment. Among the 60 samples collected based on a visually blood engorged state, we identified 45 samples from which we could recover between 1 and 11 contigs with *Vertebrata* LCAs. While the read depth coverage of those contigs varied, the majority of samples contained contigs supported by >500 reads. We also found that many of the contigs supporting bloodmeal calls were highly homologous between different samples, consistent with multiple mosquitos having fed on the same species of blood meal.

Are there really no contigs with an LCA in vertebrates in the non blood-fed mosquitoes?

After excluding any samples with LCAs corresponding to the order Primates (because of the possibility of contamination with human DNA), we identified 1 additional sample among the non blood-fed mosquitoes with a clear signal for a *Vertebrata* LCA. This corresponded to sample CMS002_025c, from which we recovered *Rodentia* contigs corresponding to *Mus musculus* call.

Also, what do we think happened in the mosquitoes that were visibly bloodfed but nothing was found; any speculation?

This could be a result of a number of different factors. We have added a final paragraph to the Blood meal sources and xenosurveillance section of the Discussion to address these issues and raise a possible strategy to enhance blood meal host recovery via RNA-based mNGS:

“Detection of a potential blood meal host in 45/60 of the blood engorged mosquitoes via RNA-based mNGS analysis is encouraging, especially in light of additional evidence of accompanying microbial cargo. The lack of success across all blood engorged mosquitoes may reflect factors that contribute to insufficient blood meal host RNA in the final extracted samples, such as variability in actual blood engorgement between mosquitoes, insufficient amounts of RNA circulating in the blood of certain types of animal hosts, or degradation/digestion of the blood meal RNA within the mosquito prior to extraction. Second, there may be too little information content in the RNA present, that is if it is from a highly conserved portion of a universal gene, there may not be an LCA below “cellular organisms” or “eukaryota”. A hybrid approach, in which primers for the enrichment of conserved, highly expressed “barcode” genes described above are incorporated at the reverse transcription step (similar to metagenomic sequencing with spiked primer enrichment strategy (MSSPE) that has been applied to enrich specific viral species (Deng et al., 2020)) may provide a path forward to boost both the recovery rate and resolution of blood meal host identification in the context of RNA-based mNGS analyses.”

3. The study of co-occurrence, although not novel, is a nice addition to the mosquito virome/microbiome determination field. Identifying novel segments and missed segments of viruses is very nice. I do however wonder: did it ever occur that co-occurrence finds a 'linked' fragment that was clearly wrong? Were some post-analyses done to check if the results make sense? It seems, especially because the paper elaborates on examples, that you need some follow-up. This is not problematic, but a nice addition to the paper would be (as is also described below) to mention which segments were added to viral genomes by co-occurrence and if some checks were done to verify these hits.4. Being able to say something about differences in viruses within the same mosquito species is super interesting. Pools do not give the possibility to say something about profiles and prevalence and the large size (148 mosquitoes) allows to find interesting correlations.What parts do you think are problematic?1. We question the validity 'blood-meal calls' as outlined above.

Reviewer 1 raised several questions and comments related to the blood meal analysis that we have addressed in-line above.

2. In this study they use % of non-host reads as a measure for the abundance of a pathogen (see e.g. Figure 3). I don't understand this at all… If you have more pathogens, then the amount of non-host reads would have to go up right? It seems to assume that the amount of non-host reads you have is similar in all samples? It becomes even more problematic when the trend is mentioned that having a higher % of non-host reads for Wolbachia is related to a lower % of non-host reads for viruses. This seems to be trivial as the amount of non-host reads goes up with increased Wolbachia infection, and therefore the % of non-host reads for viruses goes down due to the larger denominator. A different number than 'non-host reads' should be taken to normalise the data and say something about abundance. E.g. host reads or spiked RNA?

We thank Reviewer 1 for sharing these detailed and constructive comments on the lack of clarity related to the non-host read metrics we utilized in the manuscript. We address their larger question on the choice of this metric below, then describe the revisions we have made to further clarify this and address the additional points they have raised:

– In this study they use % of non-host reads as a measure for the abundance of a pathogen (see e.g. Figure 3). I don't understand this at all…

A key point of the analysis summarized in Figure 3 was to explore and compare the relative proportion of viruses, selected prokaryotic and eukaryotic microbes within and across individual mosquitoes.

– If you have more pathogens, then the amount of non-host reads would have to go up right?

Yes that is correct and can be seen in Figure 2—figure supplement 2.

However, in Figure 3, our main aim is to explore the composition of the non-host sequence space among the individual mosquitoes. We were particularly interested in elucidating the degree to which the relative proportion of non-host microbes varied across individuals, species, and collection locations. In an effort to address the concerns and questions from Reviewer 1 on this section, we have revised the Results section related to Figure 3 to better articulate this. We have also included references in this section to Figure 2—figure supplement 2 which shows 3 complementary views of per-mosquito distribution of non-host reads that we hope helps to provide further clarifying context for the reader:

1. Total unassembled non-host reads colored by whether they were assembled into contigs or not and their corresponding high-level taxonomic assignments [various colors]

2. The percentage of non-host reads assembled into contigs and assigned to high level taxonomic bins expressed as a fraction of the total reads acquired from each mosquito

3. The proportion of non-host reads assembled into contigs and assigned to high level taxonomic bins expressed as a fraction of the total number of non-host reads assembled into contigs.

– It seems to assume that the amount of non-host reads you have is similar in all samples?

No, this is not our assumption. We presented these data as normalized to total non-host reads because we (a) recognize that the amount of non-host reads recovered varies across individual mosquitoes, and (b) our focus was to explore the variability in the proportion of viruses, bacteria, and eukaryotic microbe sequences identified within the non-host read fraction of individual mosquitoes.

– It becomes even more problematic when the trend is mentioned that having a higher % of non-host reads for Wolbachia is related to a lower % of non-host reads for viruses. This seems to be trivial as the amount of non-host reads goes up with increased Wolbachia infection, and therefore the % of non-host reads for viruses goes down due to the larger denominator.

We thank Reviewer 1 for pointing out the problematic nature of our statements related to trends in the fractional changes in Wolbachia and virus where both are present. We acknowledge this point should be removed from the text of Exploring the impact of Wolbachia endosymbionts in the Results section. We have revised to highlight a distinct observation: a detectable shift in the distribution of different types or numbers of viruses detected in two mosquito species based on the per-mosquito number of Wolbachia reads. We have also included an additional Figure 3—figure supplement 1 to quantify this in more detail, and have revised comments on the Wolbachia observations in the Discussion section accordingly.

– A different number than 'non-host reads' should be taken to normalise the data and say something about abundance. E.g. host reads or spiked RNA?

We respectfully disagree. It is our hope that, taken together, our clarification above and the aforementioned revisions of the text associated with Figure 3, an explicit reference pointer to Figure 2—figure supplement 2 in the text associated with Figure 3, and the addition of Figure 3—figure supplement 1 address this point.

3. The study of co-occurrence, although not novel, is a nice addition to the mosquito virome/microbiome determination field. Identifying novel segments and missed segments of viruses is very nice. I do however wonder: did it ever occur that co-occurrence finds a 'linked' fragment that was clearly wrong? Were some post-analyses done to check if the results make sense? It seems, especially because the paper elaborates on examples, that you need some follow-up. This is not problematic, but a nice addition to the paper would be (as is also described below) to mention which segments were added to viral genomes by co-occurrence and if some checks were done to verify these hits.

We thank Reviewer 1 for this helpful feedback. We have revised the Results section to include an additional “Validation of dark segments” to address the questions raised here. Further, upon review of our BLAST results we found that some segments identified as “dark” actually did have BLAST hits (*Aedes aegypti* totivirus and Nuyav virus), and the sole “dark” segment left (other than overwhelmingly supported *Orthomyxoviridae* or *Narnaviridae* segments) was RNA 8 of Elemess virus (*Reoviridae*). We have updated all figures and data files to reflect this. We also lay out a modest case for why we believe that numerous contigs co-occurring with Elemess virus which we called RNA 8 are genuine segments in the “Validation of dark segments” paragraph.

What are the most relevant questions you are left with?1. I am curious about the limited overlap with Sadeghi et al., 2018, who sequenced so many Culex mosquitoes in California. I would suggest to say a little but more about these discrepancies and their potential causes in the discussion.

This is addressed in the Discussion section Distribution of microbes within mosquito populations, as follows: “In contrast, 10 of the 24 previously described viruses we recovered in this study correspond to viral agents described in a recent bulk mNGS of approximately 12,000 mosquitoes collected in California in 2016 (Table 1, Sadeghi et al., 2017) or an mNGS study of 7 single mosquitoes collected in 2013 (Chandler et al., 2015). The remaining 15 previously described viral genomes have been observed outside of the state of California, in some cases on completely different continents. While technical differences in study design, types of mosquito populations examined, sample processing prior to RNA extraction, and sequencing approaches likely play a major role in the distinct set of viral genomes we report here, it is also likely that additional broader factors related to less well-understood aspects of the wild-caught mosquito virome and disease ecology-- mainly the variability in the prevalence and distribution of mosquito viruses across locations, time and mosquito host species -- form the basis for these observations.”

2. What do the authors think are in those 'dark reads'? Is the amount of dark reads the same across the different samples? Similarly, are the 'tetrapoda' reads reduced/absent in mosquitoes with a reference genome available?

For the first question: As these correspond to reads assembled into contigs with no discernable sequence similarity to any published sequences via BLAST, it is very hard to say beyond “dark”. It’s clear that the vast majority of contigs are unlikely to be coding (the longest ORF in more than 50% of contigs are <64 amino acids long and in 95% the longest ORF is <102 amino acids) and tend to be poorly supported (more than 50% of contigs have <7 reads to support them and more than 95% have <24 reads). While mining these dark contigs for meaningful data is a departure from the primary goal of our study, we can say that cursory searches with HHpred amongst the longest ORFs in dark contigs covered with highest numbers of reads tend to indicate that dark contigs appear to be a mix of potentially viral and eukaryotic genes, albeit with extremely low E-values.

For the second question: the amount of “dark” reads varies across the different samples; the data related to this are available in contigs.stats.all.tsv. Reads in “dark contigs” as a proportion of all reads in contigs range from 0.001 to 0.957 across samples, with a mean of 0.292 and standard deviation of 0.260.

For the third question: Yes, we observe a trend of slightly higher yield of *Hexapoda* reads remaining in the initial non-host reads output from IDseq among samples derived from mosquito species without a reference genome. This is most pronounced for the more divergent *Culiseta* species. On average, > 0.2 of the total *Hexapoda* reads in among the *Culiseta* samples were identified as *Hexapoda* after the IDseq host filtering steps. In contrast, in *Aedes aegypti*, *Aedes albopictus* and *Culex quinquefasciatus* samples, on average < 0.05 of the total *Hexapoda* reads were identified after the initial IDseq host filtering steps.

3. In the first part of the results, mention is made to being able to characterize to kingdom level 77% of the 13 million non-host reads (also see comment on non-host reads below). I am however puzzled with the description in the text and supplemental figure 3: which 3 million contigs were not able to be characterized? Where in supplemental figure 3 are they? This is especially puzzling as the main text mentions that 11 million non-host reads are from complete viral genomes, 0.9 million to eukaryotic taxa and 0.7 million to prokaryotic taxa?

We thank Reviewer 1 for pointing out a confusing typo in the Results section this should have read “…77% of the 21.8 million non-host reads assembled into contigs with > 2 reads” and we have corrected this in the revised version. This refers to the 13M reads assembled into contigs. We have corrected this error and reviewed all read numbers summarized in the text of the Results section and in the more detailed data outlined in Figure 2—figure supplement 1 to ensure they are harmonized throughout the manuscript.

4. There seem to be 131 bars, corresponding to individual mosquitoes, in figure 3? Where arethe remaining 17?

The remaining 17 samples fell into 1 of 3 possible categories: (1) they did not harbor viral contigs from family categories shown or Wolbachia contigs supported by >1% non-host reads, (2) they did not harbor Wolbachia, Trypanosomatidae, Apicomplexa, or Nematoda contigs supported by < 1% of non-host reads, or (3) they fell into both of the above categories.

What are your tips (in addition to responses to above questions)?1. I think the definition of 'non-host reads' needs to be clearly made and used consistently across the document. At the end of the paragraph 'Comprehensive and quantitative analysis of non-host sequences detected in single mosquitoes' the concept of "…13 million non-host reads…" is introduced. At first glance of supplemental figure 3 it seems that "non-host reads" could also be defined as the 16.7 aligned reads that are left after putative host sequences are removed. Although it is true that the derivation of 13 million is explained in the figure text of supplemental figure 3, it may be easier for the reader (as it cost me some time) to explain this in the main text. In addition, is the definition of 'non-host reads' (corresponding to 13-million reads) corresponding to "classified non-host reads" in the following excerpt: "For every sample, "classified non-host reads" refer to those reads mapping to contigs that pass the above filtering, Hexapoda exclusion, and decontamination steps. "Non-host reads" refers to the classified non-host reads plus the reads passing host filtering which failed to assemble into contigs or assembled into a contig with only two reads."? This caused some confusion.

We thank Reviewer 1 for these helpful comments. We have revised the Materials and methods section to streamline the read on contig/read filtering and nomenclature, and harmonized the terms for different classes of non-host reads throughout the manuscript to minimize this sort of confusion in the revised version.

2. I believe it would be a valuable addition to add a table for the viruses which includes: (1) How it was determined that the complete genome is there, (2) The percentage overlap for those segments that were identified with blast and (3) Which viruses were already known.

1. Where possible, complete viral genome recovery was inferred via identification of canonical (for viral family or genus) gene and segment presence/absence in assembled contigs. We have summarized this explicitly in the final paragraph under Final Classification in the Materials and methods section. As essentially the same method was applied across all candidate viral genome sequences, we do not see that it merits an additional column in Table 1 or Data S4a/b.

2. We have revised supplemental data to accommodate this information as follows: Data

S4 file has been renamed to Table 1—source data 1 and a new Table 1—source data 2, that

includes the requested information for viral segments identified by blast alignment has been added.

3. All rows with value “FALSE” under column header “novel?” in Table 1 correspond to viruses identified in the study that were already known.

3. Have the numbers of the caught mosquitoes somewhere written out in the Materials and methods.

We revised the Materials and methods section to include the number of mosquitoes and a reference to Figure 1—source data 1 (mosquito demographics metadata) in the Materials and methods section to address this suggestion.

4. Pg2 L1-3: "Metagenomic sequencing….. a single assay." Perhaps a bit early for this statement. Would suggest to place it two paragraphs later before:"Here, we analyzed…."

We agree and have revised this sentence as follows:

Original sentence:

“Metagenomic sequencing of individual mosquitoes provides a means to comprehensively identify mosquito species, the pathogens they carry and the animal hosts that define a transmission cycle with a single assay.”

Revised sentence:

“Metagenomic sequencing of individual mosquitoes offers a potential single assay to comprehensively identify mosquito species, the pathogens they carry, and the animal hosts that define a transmission cycle.”

5. Figure S4 is too pixelated to read. Perhaps due to pdf conversion, but please do check before submission.

We can provide separate pdfs for all main and supplemental figures for the manuscript submission. These and all Supplemental Data files have also been uploaded separately to a FigShare repository we have established or the manuscript (https://doi.org/10.6084/m9.figshare.11832999.v3).

Reviewer #2 (Evidence, reproducibility and clarity (Required)):Summary:In this study, the authors utilized unbiased meta-transcriptomic in sequencing 148 diverse wild-caught mosquitoes (Aedes, Culex, and Culiseta mosquito species) collected in California, with main aim of detecting sequences of eukaryotic, prokaryotic and viral origin. Their results show that majority of their sequenced data assembled into contigs corresponding to viral genomes. In their data, 7.4 million viral reads clustered as +ssRNA viruses including Solemoviridae, Luteoviridae, Tombusviridae, Narnaviridae, Flaviviridae, Virgaviridae, and Filovirida whereas 2.25 million viral reads identified as -ssRNA viruses comprising of Peribunayviridae, Phasmaviridae, Phenuiviridae, Orthomyxoviridae, Chuviridae, Rhabdoviridae, and Ximnoviridae. With 0.94 million viral reads, dsRNA viruses formed the third most abundant virus category with viruses under families Chrysoviridae, Totiviridae, Partitiviridae, and Reoviridae. Under the prokaryotic taxa, Wolbachia species was the dominant group, followed by other lower abundance bacterial taxa that includes Alphaproteobacteria, Gammaproteobacteria, Terrabacteria group, and Spirochaetes. Trypanosomatidae was the most dominant eukaryotic taxa, followed up by reads from Bilateria and Ecdysozoa taxa. Ultimately, this study demonstrates that single mosquito meta-transcriptomic analysis has potential in identifying vectors of human health significance, potent emerging pathogens being transmitted by them and their reservoirs all in one assay.Major comments:1. Are the key conclusions convincing? The conclusions are accurate.2. Should the authors qualify some of their claims as preliminary or speculative, or remove them altogether?None. The study's results, discussion and conclusion are appropriate.3. Would additional experiments be essential to support the claims of the paper? Request additional experiments only where necessary for the paper as it is, and do not ask authors to open new lines of experimentation.As much as the authors describe the use of mNGS as a tool in validating mosquito species and providing an unbiased look at the vector-associated pathogens, it is still prudent for them to use qPCR to validate the obtained RNASeq data (e.g. validation of the viral sequences).

Given qPCR generally targets small (~100bp) sequences equivalent to a single read in our data, and the fact that the bulk of the viral genomes in our study were derived from larger, assembled contigs supported by hundreds, and in some cases, thousands of reads, we do not see that these suggested experiments are merited.

4. Are the suggested experiments realistic in terms of time and resources? It would help if you could add an estimated cost and time investment for substantial experiments. The outlined methodology is realistic.

The limited amount of remaining RNA from each of the wild-caught mosquitoes presents a major obstacle to accomplishing such analyses. Moreover, with 70 viral agents, the suggested qPCR assays would entail design and validation of a multiple PCR primers and synthesis of a battery of positive controls for each of the detected viruses.

5. Are the data and the methods presented in such a way that they can be reproduced?The methodology is reproducible.6. Are the experiments adequately replicated and statistical analysis adequate?YesMinor comments:1. Specific experimental issues that are easily addressable. qPCR validation the obtained RNASeq data should be conducted.

See our response above to this suggestion in Major Comments, items 3 & 4.

2. Are prior studies referenced appropriately?The recently publications about mosquito microbiome/virome should be added. (eg. doi:10.1128/mSystems.00640-20.)

We thank Reviewer 2 for the suggestion and have updated our citations and bibliography to incorporate this reference into the introduction of the manuscript.

3. Are the text and figures clear and accurate?The resolution for Figure 4, Figure 6, SFigure 2, SFigure 4, and SFigure 5 is poor. The author should update them.

We can provide separate pdfs for all main and supplemental figures for the manuscript submission. These and all Supplemental Data files have also been uploaded separately to a FigShare repository we have established or the manuscript (https://doi.org/10.6084/m9.figshare.11832999.v3).

4. Do you have suggestions that would help the authors improve the presentation of their data and conclusions?In the method section, the mosquito has been washed to avoid the contamination from the environment before RNA extraction?

The mosquitoes included in this study were not washed prior to extraction.

We have altered the text to acknowledge the possibility of externally sourced microbes in the Results section, final paragraph of Diverse known and novel RNA virus taxa dominate the mosquito microbiota:

“We cannot rule out that the known and novel viral species that correspond to viral families previously thought to only infect plants and fungi, (e.g., the *Chrysoviridae*, *Totiviridae*, *Luteoviridae* and *Solemoviridae*, Table 1) could potentially be explained by an environmental exposure retained on the surface of the mosquito.”

We have also addressed this limitation of our study more broadly in Distribution of microbes within mosquito populations in the Discussion section:

“Intriguingly, a significant fraction (n=46) of the 70 viral genomes we identified in this study correspond to novel divergent viruses. Further studies will be required to understand each of these viruses, and whether they correspond solely to microbial cargo (i.e., non-infecting viral species hitching a ride on the mosquito exterior or via ingestion of blood or nectar), versus insect specific viruses, or potentially transmissible human or animal pathogens.”

2. Most part of non-host reads are matched to the viruses (10.5M), however only few of them were belong to the prokaryotes, does it means mosquito carries more viruses than prokaryotes.

In aggregate, among the non-host reads that could be assembled into contigs and taxonomically assigned, viruses predominated. These results are summarized in Figure 2 and Figure 2—figure supplement 3. At the individual level, as shown in Figure 3 and Figure 3—figure supplement 1, there are examples where we can see the prokaryote Wolbachia making up the majority of non-host reads assembled into contigs (see especially the Culex pipiens mosquitoes).

3. None of the mosquito-borne virus known to occur in California (eg. WNV, SLEV, WEEV, ) has been found in Table 1 for the virus detected with complete genome in this study. In contigs level, did the author detected any mosquito-borne virus known to occur in California. Since the mNGS is very sensitive and this study include large sample numbers, why no known mosquito-borne virus was detected in their study should be discussed.

Despite the sensitivity of mNGS neither partial nor complete genome sequences from WNV, SLEV, or WEEV were recovered in this study. We have aggregated data related to the number of reported WNV and SLEV positive mosquito pools at each site based on the California Arbobulletin and compared dates of positive mosquito pools to our sample collection dates (see Data S21). In short, WNV and SLEV were reported at low frequency and in only 2 of the collection sites during the study period (Placer and San Diego; range 0-10 mosquito pools tested positive for WNV, and 0-1 pools positive for SLEV). Seven samples from Placer were collected on 11/21/17 and 11/28/17 and overlapped with a reporting period spanning 11/18/17-12/1/17 during which a single mosquito pool was found positive for SLEV. We do not have resolution on dates for when that positive pool was identified. For San Diego, the collection dates for samples included in the study did not overlap with any of the reporting periods in which a mosquito pool tested positive for WNV or SLEV. We do acknowledge that Reviewer 2 raises an important question to be addressed and have added the text below under Distribution of microbes within mosquito populations the Discussion section:

“The lack of detection of these arboviruses likely reflects the low rate of circulation in the participating control districts during the sample collection period for the study. Indeed, both viruses were detected only sporadically and in only 2 of the 5 participating sites (see Supplemental Data File 21 for an alignment of data from reports for mosquito pools testing positive for WNV and SLEV and sample collection statistics).”

Reviewer #2 (Significance (Required)):1. Describe the nature and significance of the advance (e.g. conceptual, technical, clinical) for the field.With the existential threat of emerging novel pathogens of global health concern, efficient and rapid public health surveillance strategies are crucial in monitoring and possibly averting such eventual calamities. Specifically, mosquitoes are widely diverse and are known to harbor and transmit various pathogenic agents to humans and animals. Thus, this rapid identification of relevant vector species, pathogens and their reservoirs in one assay is a promising and convenient aspect of surveillance in the public health sector.2. Place the work in the context of the existing literature (provide references, where appropriate). Shi et al. reported the first single mosquito viral metagenomics study, in which her and the team demonstrated the feasibility of using single mosquito for viral metagenomics, a methodology that has potential to provide much more precise virome profiles of mosquito populations. In the present study, the authors have gone a step higher by aiming to combine three objective points in single mosquito meta-transcriptomic, as described in brief in their abstract and the comprehensive methodology outline.Reference: Shi, C., Beller, L., Deboutte, W. et al. Stable distinct core eukaryotic viromes in different mosquito species from Guadeloupe, using single mosquito viral metagenomics. Microbiome 7, 121 (2019). https://doi.org/10.1186/s40168-019-0734-23. State what audience might be interested in and influenced by the reported findings. The methodology and findings described in this manuscript are important in advancing the public health field of vector surveillance. The identification of relevant vector species, pathogens and their reservoirs in one assay is a promising and convenient aspect of surveillance.4. Define your field of expertise with a few keywords to help the authors contextualize your point of view. Indicate if there are any parts of the paper that you do not have sufficient expertise to evaluate.I am an Associate Professor at a research institute. My lab research work focuses onArbovirology studies, more specifically vector surveillance of known and novel viruses associated with mosquitoes and ticks, mosquito-transcriptomic studies, mosquito viruses tropism studies and other related mosquito-virus interaction studies.Reviewer #3 (Evidence, reproducibility and clarity (Required)):Summary:The authors demonstrate a powerful method utilizing mNGS of individual mosquitoes utilizing reference-free analysis. This allows researchers to combine the resulting datasets of mosquito identification, blood-meal source, microbiome, viral sequencing, etc. Such knowledge could be a useful tool in detecting and responding to transmission of mosquito-borne diseases that affect human or animal populations, even though the technology is currently likely too expensive for widespread use (as acknowledged by the authors).Major Comments:No major revisions requested.The authors provide their detailed methodology, including code, allowing for replication by other groups.Minor Comments:The authors' discussion of using this technique in order to detect pathogens should be qualified regarding detection vs possible transmission. Detecting a virus in an engorged mosquito does not necessarily mean that said mosquito can transmit the virus, but may have simply acquired it from a recent blood meal. The same can be said of detecting a plant pathogen following a recent sugar meal.

We agree with Reviewer 3, and have altered the text in the Discussion section Blood meal and xenosurveilance to better highlight this point:

“This sort of novel virus-blood meal host observation provides a starting point to understanding of the interplay between viruses and microbes circulating within the animal and insect populations in a given location. However, directed analyses in the lab and the field would be required to establish if such linkages (a) indeed represent microbes residing in the putative animal host populations, and (b) those microbes can be taken up via blood meal, and maintained and transmitted by mosquitoes”

From the methods, it seems that mosquitoes were not washed prior to processing. This may make it difficult to discriminate between internal and external microbiota as well as lead to cross-contamination of surface microbiota between mosquitoes collected in the same trap.

Reviewer #3 is correct, the mosquitoes sequenced in this study were not washed prior to RNA extraction. We have altered the text to acknowledge the possibility of externally sourced microbes in the Results section, final paragraph of Diverse known and novel RNA virus taxa dominate the mosquito microbiota:

“We cannot rule out that the known and novel viral species that correspond to viral families previously thought to only infect plants and fungi, (e.g., the Chrysoviridae, Totiviridae, Luteoviridae and Solemoviridae, Table 1) could potentially be explained by an environmental exposure retained on the surface of the mosquito.”

We have also addressed this limitation of our study more broadly in Distribution of microbes within mosquito populations in the Discussion section:

“Intriguingly, a significant fraction (n=46) of the 70 viral genomes we identified in this study correspond to novel divergent viruses. Further studies will be required to understand each of these viruses, and whether they correspond solely to microbial cargo (i.e., non-infecting viral species hitching a ride on the mosquito exterior or via ingestion of blood or nectar), versus insect specific viruses, or potentially transmissible human or animal pathogens.”

Reviewer #3 (Significance (Required)):This work currently would be of interest to other research groups examining the co-occurence of pathogens, other microbiota, and blood meals for field collected mosquitoes. While of great potential application to public health surveillance, the current cost is likely prohibitive.My field of expertise is virology and vector biology with minimal background in NGS.